# Test-Time Adaptation via Conjugate Pseudo-labels

**Sachin Goyal**[*1]     **Mingjie Sun**[*1]     **Aditi Raghunathan**[1]     **Zico Kolter**[1,2]

[1]Carnegie Mellon University, [2]Bosch Center for AI

{sachingo, mingjies, raditi, zkolter}@cs.cmu.edu

## Abstract

Test-time adaptation (TTA) refers to adapting neural networks to distribution shifts, with access to only the unlabeled test samples from the new domain at test-time. Prior TTA methods optimize over unsupervised objectives such as the entropy of model predictions in TENT [50], but it is unclear what exactly makes a good TTA loss. In this paper, we start by presenting a surprising phenomenon: if we attempt to *meta-learn* the "best" possible TTA loss over a wide class of functions, then we recover a function that is *remarkably* similar to (a temperature-scaled version of) the softmax-entropy employed by TENT. This only holds, however, if the classifier we are adapting is trained via cross-entropy loss; if the classifier is trained via squared loss, a different "best" TTA loss emerges. To explain this phenomenon, we analyze test-time adaptation through the lens of the training losses's *convex conjugate*. We show that under natural conditions, this (unsupervised) conjugate function can be viewed as a good local approximation to the original supervised loss and indeed, it recovers the "best" losses found by meta-learning. This leads to a generic recipe that can be used to find a good TTA loss for *any* given supervised training loss function of a general class. Empirically, our approach consistently dominates other TTA alternatives over a wide range of domain adaptation benchmarks. Our approach is particularly of interest when applied to classifiers trained with *novel* loss functions, e.g., the recently-proposed PolyLoss [25] function, where it differs substantially from (and outperforms) an entropy-based loss. Further, we show that our conjugate based approach can also be interpreted as a kind of self-training using a very specific soft label, which we refer to as the *conjugate pseudo-label*. Overall, our method provides a broad framework for better understanding and improving test-time adaptation. Code is available at https://github.com/locuslab/tta_conjugate.

## 1   Introduction

Modern deep networks perform exceeding well on new test inputs that are close to the training distribution. However, this performance dramatically decreases on test inputs drawn from a different distribution. While there is a large body of work on improving the robustness of models, most robust training methods are highly specialized to the setting they cater to. For e.g., they assume pre-specified perturbations, subpopulations, and spurious correlations, or access to unlabeled data from the target distribution, and most methods offer close to no improvement on general distribution shifts beyond what they were trained for [12, 21].

In practice, it is often cumbersome (or even impossible) to precisely characterize all possible distribution shifts a model could encounter and then train accordingly. Instead, a model already trained on some source data must be able to *adapt* at test-time to new inputs from a different domain. This setting of *test-time adaptation* (TTA) has gained interest in recent years [6, 47, 50, 54]. TTA is typically accomplished by updating the source model parameters via a few steps of optimization on an *unsupervised objective* involving the new test sample from the target distribution. The choice

---

[*] Equal Contribution

36th Conference on Neural Information Processing Systems (NeurIPS 2022).

of this unsupervised objective, which we call the TTA loss, dictates the success of the adaptation procedure. [47] uses a self-supervised objective on the test sample, [50] uses the entropy of model predictions, and several follow-ups have proposed variants or alternatives [40, 54]. However, it remains unclear as to how to choose or guide the selection of this TTA loss, and thus far the choice of these losses has remained largely heuristic in nature.

In this work, we begin by presenting a set of intriguing experiments where we attempt to *learn* the "best" TTA loss for a given source classifier and distribution shift. We parameterize the TTA loss by another neural network whose parameters are learnt via meta-learning [3, 9] where we differentiate through the adaptation process to find the TTA loss that achieves the best adaptation on distribution shifts. Surprisingly, we ultimately learn a TTA loss that looks *remarkably* similar to (a temperature-scaled version of) the softmax-entropy loss, which was already proposed by [50]. Why did we recover the commonly used softmax-entropy loss despite the fact that the procedure is capable of learning a very general class of losses and the meta-learning process could potentially *specialize* to both the source classifier and the distribution shift of interest? Furthermore, we find that this pattern only holds when the loss used to train the source classifier is cross-entropy loss; when a different loss such as squared loss is used instead, the meta-learning procedure recovers a TTA loss that itself looks more like a negative squared error, and is very different from the softmax-entropy loss (Section 3).

In order to explain this phenomenon, we propose to consider TTA through the lens of the *convex conjugate* function. Specifically, given a hypothesis function $h(x)$ and label $y$, several common losses (cross-entropy and the squared loss amongst them, but not limited to these) can be written in the form $\mathcal{L}(h(x), y) = f(h(x)) - y^T h(x)$ for some function $f$. In these cases, we show that "natural" TTA loss for such classifiers is precisely the (negation of) the convex conjugate evaluated at the gradient of $h$, $\mathcal{L}_{\text{TTA}}(x) = -f^*(\nabla f(h(x)))$, where $f^*$ is the convex conjugate of $f$. This framework not only recovers the results of our meta-learning experiments, but also justifies why some specific choices of TTA loss in the previous literature work well (e.g., this framework recovers TENT's choice of softmax-entropy for cross-entropy-trained classifier). Moreover, it also provides a broad framework for what the TTA loss should be when the source model is trained using various different loss functions (for example the recently-proposed PolyLoss [25, 29]) as is becoming increasingly common in machine learning. Further, we show that our proposed conjugate adaptation loss is in fact a kind of self-training with pseudo-labels [42], a classic approach in machine learning. Various formulations of the pseudo-label have been proposed in the literature, and our conjugate analysis provides a general recipe for the "correct" choice of soft pseudo-labels given by $\hat{y}(x) = \nabla f(h(x))$. We thus refer to these as *conjugate pseudo-labels* (Conjugate PL's), and believe our work provides a broad framework for understanding adaptation with unlabeled data in general.

Finally, we empirically verify the effectiveness of our proposed conjugate adaptation loss across several datasets and training losses, such as cross-entropy and squared loss, along with the recently-proposed PolyLoss [25] (which itself has shown higher standard test accuracy on a wide range of vision tasks). Over *all* models, datasets and training losses, we find our proposed conjugate pseudo-labeling consistently outperforms prior TTA losses and improves TTA performance over the current state of the art.

## 2   Background and preliminaries.

**Test-time adaptation.** We are interested in mapping an input $x \in \mathbb{R}^d$ to a label $y \in \mathcal{Y}$. We learn a model $h_\theta : \mathbb{R}^d \mapsto \mathbb{R}^{|\mathcal{Y}|}$ parameterized by $\theta$ that maps an input $x$ to predictions $h_\theta(x)$. We assume access to a trained source model and adapt at test-time over the test input, before making the final prediction. This is the standard test-time adaptation (TTA) setting [47, 50]. During TTA, we update the model parameters on an unsupervised objective $\mathcal{L}(x, h_\theta)$. For example, in TENT [50], this loss is the entropy of the softmax-normalized predictions of the model. At each time step of adaptation, we observe a batch of test inputs and we take a gradient step towards optimizing the TTA loss on this test batch. As is standard, we measure the average online performance of models across all steps (number of test batch inputs seen) in the adaptation process.

**Meta learning the loss function.** In order to explore the existence of different TTA losses, we employ the meta-learning procedure where we attempt to *learn* the TTA loss. We use a similar procedure as prior work on meta-learning loss functions [3, 37] and parameterize the loss function via a neural network $m_\phi : \mathbb{R}^{|\mathcal{Y}|} \mapsto \mathbb{R}$ that takes in the model predictions/logits and outputs a loss value. We want to learn parameter $\phi$ such that when we update $\theta$ via the loss function $m_\phi$, our final

performance is optimal. In order to do so, let $x$ be the unlabeled test samples to adapt to, and $y$ be the corresponding labels. We update $\theta$ and $\phi$ alternatively as follows.

$$\theta^{t+1} \leftarrow \theta^t - \alpha \frac{\partial m_{\phi^t}(h_{\theta^t}(x))}{\partial \theta^t}, \quad \phi^{t+1} \leftarrow \phi^t - \beta \frac{\partial \mathcal{L}(h_{\theta^{t+1}}(x'), y')}{\partial \phi^t}, \tag{1}$$

where $\mathcal{L}$ is some supervised surrogate loss function such as cross-entropy. Please refer to Appendix A3 for further details regarding meta-learning setup. Note that the meta-learning process above assumes access to labels $y$ of test inputs. In this paper, we do *not* propose meta-learning the TTA loss as an approach. Rather, we use meta-learning to explore what the "best" TTA losses look like. We discuss our findings from this exploration in the next section.

## 3 Test-time Adaptation via Meta-Learnt Losses

The objective used in TENT is the softmax-entropy of the model predictions which essentially makes the classifier more confident in its current predictions. The same can be achieved by various other loss formulations such as those mentioned in [40]. With so many possible choices for the loss function, what should we use for TTA? In this section, we attempt to answer this empirically and present some intriguing observations.

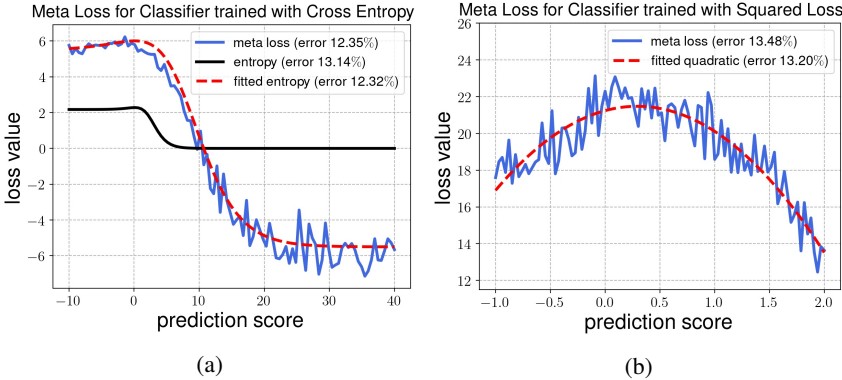

(a)                                                          (b)

Figure 1: Visualization of meta loss (blue) by varying one input prediction score. (a) For cross-entropy loss trained model, the learnt meta loss can be approximated with a scaled softmax-entropy function (dashed red). (b) When the source model is trained with a squared loss for classification, the learnt meta loss (blue) can be fitted closely with a quadratic function (dashed red), shown in Figure 1b. The range (max/min) of the prediction score (logit) in x-axis is chosen to cover the empirical range of the predicted logits.

**Experiment 1.** We learn the TTA loss parameterized by a neural network via meta-learning as described in Section 2. Our source classifier is a ResNet-26 trained on CIFAR-10 and we adapt to distribution shifts in CIFAR-10-C. We use the 4 labeled validation noises in CIFAR-10-C to learn the meta-loss network parameters and we denote the resulting learnt loss function by meta-TTA loss. We then adapt the source classifier to the test set of 15 corruptions by optimizing the meta-TTA loss.

**Observations.** First, we find that TTA using meta-TTA loss performs better than TENT ($12.35\%$ vs $13.14\%$), suggesting that there are better TTA losses than previous losses based on softmax-entropy.

However, on examining this meta-TTA loss, we find a surprising observation. Figure 1a (blue curve) visualizes the learnt meta-loss over model predictions as we vary a single class prediction with the rest fixed. Qualitatively, the learnt meta-loss looks very similar to softmax-entropy in one dimension. In fact, we can fit it closely with a scaled softmax-entropy function (dashed red curve): $\alpha \cdot \mathcal{H}(\text{softmax}(h_\theta(x)/T))$, where $\alpha$ is a magnitude parameter and $T$ is a temperature scaler. We want to test if the meta-loss is basically learning the softmax-entropy function. Hence, we perform test-time adaptation with the fitted softmax-entropy function instead (dashed red curve) and achieve an error of $12.32\%$, essentially recovering the performance of meta-TTA.

Despite the ability to represent many different loss functions and potentially specialize to the CIFAR-10-C setting, the meta-loss procedure gave back the standard entropy objective. *Do we always recover a loss that looks like softmax-entropy?*

**Experiment 2.** In an attempt to isolate when we get back the entropy objective, we vary several things. We tried different architectures for the source classifier, different losses $\mathcal{L}$ during the meta-learning process (1) and different training losses for the source classifier.

**Results.** We observed that we consistently recovered the temperature scaled softmax-entropy function in all cases *except* when we varied the training loss for the source classifier (Appendix A.10). On using the squared loss function [18], a strikingly different meta-TTA loss emerges. Figure 1b (blue curve) shows the learnt meta-loss (13.48% error) for this network. Here again, the meta-TTA loss outperforms entropy (14.57%) but it is not simply due to a scaling factor. The loss now looks like the negative squared error (red curve). Like previously, we tried fitting a quadratic loss directly to the meta loss in Figure 1b, and this time we even slightly outperformed the meta-TTA loss.

To summarize, we used a meta-learning procedure to search for the "best" TTA loss, where the loss itself was parameterized by a neural network that could potentially represent arbitrarily complex loss functions. However, we ended up with loss functions displaying remarkable structure: across different architectures and different variants of meta-learning, for a classifier trained with cross-entropy, the meta-TTA loss was temperature scaled softmax-entropy and for a classifier trained with squared loss, the meta-TTA loss was a negative squared loss. This is interesting from both a practical and conceptual standpoint where the "best" TTA loss depends on the loss used to train the source classifier in a clean fashion. We attempt to understand and explain this phenomenon in the next section.

## 4 Conjugate Pseudo Labels

Results in the previous section raise an obvious question: *why* does softmax-entropy as used in TENT seem to be the "best" possible test time adaptation loss for classifiers trained via cross-entropy (at least, best in the sense that meta-learning consistently recovers something which essentially mimics softmax-entropy, even though meta-loss is parameterized by a neural network and hence could learn much more complex functions specific to the model and the particular shift)? And why, alternatively, does a quadratic TTA loss seem to perform best when the classifier is trained via squared loss?

In this section, we offer an explanation of this phenomenon via the construct of the convex conjugate function [1]. As we will see, our method recovers softmax-entropy *and* quadratic loss as the "natural" objectives for classifiers trained via cross-entropy and squared loss respectively. Furthermore, for classifiers trained via *other* loss functions, as is becoming increasingly common in deep learning, our approach naturally suggests corresponding test-time adaptation losses, which we show in the next section to comparatively outperform alternatives. Thus, we argue that our framework overall provides a compelling recipe for specifying the "correct" method for TTA for a large class of possible losses.

### 4.1 Losses and the convex conjugate

We begin by formally considering loss functions between a hypothesis output $h_\theta(x)$ (e.g., the logit outputs of a classifier, or the direct prediction of a regressor) and target $y$ that take the following form

$$\mathcal{L}(h_\theta(x), y) = f(h_\theta(x)) - y^T h_\theta(x) \tag{2}$$

for some function $f$; when there is no risk of confusion, we will use $h$ in place of $h_\theta(x)$ for simplicity of notation. While not every loss can be expressed in such a form, this captures a wide variety of common losses (possibly scaled by a constant value). For example, cross-entropy loss corresponds to the choice $f(h) = \log \sum_i \exp(h_i)$ and where $y$ denotes a one-hot encoding of the class label; similarly, squared loss corresponds to the choice $f(h) = \frac{1}{2}\|h\|_2^2$.

When training an over-parameterized classifier, we can roughly view the training process as (approximately) attaining the *minimum* over hypotheses $h$ for each training example

$$\min_\theta \frac{1}{t} \sum_{i=1}^{t} \mathcal{L}(h_\theta(x_i), y_i) \approx \frac{1}{t} \sum_{i=1}^{t} \min_h \mathcal{L}(h, y_i) \tag{3}$$

where $t$ is the number of training samples. However, in the case of losses in the form (2), the minimization over $h$ in this form represents a very specific and well-known optimization problem: it is known as the *convex conjugate* [1] of the function $f$

$$\min_h \mathcal{L}(h, y) = \min_h \{f(h) - y^T h\} = -f^\star(y) \tag{4}$$

where $f^\star$ denotes the convex conjugate of $f$. $f^\star$ is a convex function in $y$ (and indeed, is convex regardless of whether or not $f$ is convex). Furthermore, for the case that $f$ is convex differentiable, the optimality condition of this minimization problem is given by $\nabla f(h^{\text{opt}}) = y$, so we also have that

$$f^\star(y) = f^\star(\nabla f(h^{\text{opt}})) \tag{5}$$

where $h^{\text{opt}}$ refers to the optimal classifier (used interchangeably with $h_{\theta^{\text{opt}}}$). Putting this all together, we can state (admittedly, in a rather informal manner) that under the assumption that $\theta^{\text{opt}}$ is chosen so as to approximately minimize the empirical loss on the source data in the over-parameterized setting, we have that for $t$ inputs

$$\frac{1}{t} \sum_{i=1}^t \mathcal{L}(h_{\theta^{\text{opt}}}(x_i), y_i) \approx \frac{1}{t} \sum_{i=1}^t -f^\star(\nabla f(h_{\theta^{\text{opt}}}(x_i))) \tag{6}$$

i.e., the empirical loss can be approximated by the (negative) conjugate applied to the gradient of the $f$, at least in a region close to the optimal $\theta^{\text{opt}}$ that minimizes the empirical loss. But the later expression has the notable benefit that it does not require any label $y_i$ in order to compute the loss, and thus can be used as a basis for TTA on target domain of the hypothesis function $h_{\theta^{\text{opt}}}$.

**Definition 1 (conjugate adaptation loss)** *Consider a loss function that takes the form given in 2, used for training a hypothesis $h_\theta$ in the over-parameterized regime. We define the conjugate adaptation loss $\mathcal{L}^{conj}(h_\theta(x)) : \mathbb{R}^{|\mathcal{Y}|} \mapsto \mathbb{R}$ as follows.*

$$\mathcal{L}^{conj}(h_\theta(x)) = -f^\star(\nabla f(h_\theta(x))) = f(h_\theta(x)) - \nabla f(h_\theta(x))^\top h_\theta(x). \tag{7}$$

## 4.2 Recovery of existing test-time adaptation strategies

**Cross-entropy**  The interesting aspect to this formalism is that when applied to classifiers trained with cross-entropy, it recovers exactly the TENT approach to TTA : minimizing the softmax-entropy of $h_\theta(x)$. And indeed, this loss was *also* recovered when using meta-learning to learn the "optimal" test-time adaptation loss. To see this, note that for cross-entropy, we have that $f(h) = \log \sum_i \exp(h_i)$, giving the optimality condition $y = \nabla f(h^{\text{opt}}) = \frac{\exp(h^{\text{opt}})}{\sum_i \exp(h_i^{\text{opt}})}$ and the conjugate function

$$f^\star(y) = \begin{cases} \sum_i y_i \log y_i & \text{if } \sum_i y_i = 1 \\ \infty & \text{otherwise} \end{cases}. \tag{8}$$

In other words,

$$\mathcal{L}^{\text{conj}}(h_\theta(x)) = -f^\star(\nabla f(h_\theta(x))) = -\sum_i \frac{\exp(h_i)}{\sum_j \exp(h_j)} \log \frac{\exp(h_i)}{\sum_j \exp(h_j)} \tag{9}$$

i.e. softmax-entropy of the model prediction, which is exactly the TTA loss that TENT uses.

**Squared loss**  For the squared loss, we have that $f(h) = \frac{1}{2}\|h\|_2^2$, leading to the optimality condition $y = h$ and conjugate function $f^\star(y) = \frac{1}{2}\|y\|_2^2$. Hence, the adaptation loss in this case would be simply given by $\mathcal{L}^{\text{conj}}(h_\theta(x)) = -f^\star(\nabla f(h_\theta(x))) = -\frac{1}{2}\|h\|_2^2$ which is also what we observed in the meta-learning experiments discussed in Section 3.

## 4.3 Conjugate pseudo-labels

We now emphasize that by the nature of our approximations, there is an additional simple interpretation of the conjugate loss: it is also equal to the original loss (2) applied to the "psuedo-labels" $\tilde{y}_\theta^{\text{CPL}}(x) = \nabla f(h_\theta(x))$, where CPL refers to conjugate pseudo-labels, i.e.,

$$\mathcal{L}^{\text{conj}}(h_\theta(x)) = -f^\star(\nabla f(h_\theta(x))) = f(h_\theta(x)) - \nabla f(h_\theta(x))^T h_\theta(x) = \mathcal{L}(h_\theta(x), \nabla f(h_\theta(x))). \tag{10}$$

This property is known as the Fenchel-Young inequality, that is $f(x) + f^\star(u) \geq x^T u$ holding with equality when $u = \nabla f(x)$. In other words, our conjugate adaptation loss is precisely equivalent to self-training under the specific soft pseudo-labels given by $\tilde{y}^{\text{CPL}} = \nabla f(h_\theta(x))$. And indeed, for many cases, this may be a more convenient form to compute than explicitly computing the conjugate function at all. For this reason, we refer to our method as that of *conjugate pseudo-labels*.

In the case of cross-entropy loss, this approach then corresponds exactly to self-training using labels given by the softmax applied to the current hypothesis. We must emphasize, however, that while our conjugate formulation indeed has this "simple" form for the case of cross-entropy loss, the real advantage comes in that it provides the "correct" pseudo-label for use with *other* losses, which may result in pseudo-labels different from the "common" softmax operation.

**Example: conjugate pseudo-labels for PolyLoss.** PolyLoss [25] is a recently-proposed simple alternative to cross-entropy loss than has been shown to improve performance across a wide variety of compute tasks. This loss is given by the form

$$\mathcal{L}_{\text{poly}}(h_\theta(x), y) = \mathcal{L}_{\text{ce}}(h_\theta(x), y) + \epsilon \cdot y^T(1 - \text{softmax}(h_\theta(x))) \tag{11}$$

We note that this can be put exactly into our conjugate form (equation 2) by writing the loss in a slightly more involved fashion, which we refer to as the *expanded conjugate form*

$$\mathcal{L}_{\text{poly}}(h_\theta(x), y) = f(h_\theta(x)) - y^T g(h_\theta(x)). \tag{12}$$

where $f$ is the log-sum-exp function as before, and $g(h) = h - \epsilon(1 - \text{softmax}(h))$. In order to formally put this into the form of the previous loss function (equation 2), we can simply define an alternative hypothesis as the function $h'_\theta(x) = g(h_\theta(x))$, and then define PolyLoss in the conjugate form as

$$\mathcal{L}_{\text{poly}}(h'_\theta(x), y) = f(g^{-1}(h'_\theta(x))) - y^T h'_\theta(x). \tag{13}$$

Typically, however, it is easier to simply operate on the expanded conjugate form, which yields the optimality condition for the pseudo-label $\nabla f(h^{\text{opt}}) = \mathrm{D}g(h^{\text{opt}})\tilde{y}_\theta^{\text{CPL}}(x)$, where D is the Jacobian operator. For the case of PolyLoss, this leads to the conjugate pseudo-label of the following form: $\tilde{y}_\theta^{\text{CPL}}(x) = (I + \epsilon\,\text{diag}(z) - \epsilon z z^T)^{-1} z, \ \ z \equiv \text{softmax}(h_\theta(x))$.

**Test-time adaptation.** Finally, we note that the above discussion doesn't actually address any topics related to test-time adaptation to OOD data, but merely provides a generic characterization of a self-training procedure for generic loss functions of the form (2). However, the application to TTA on OOD data is fairly straightforward: as long as the learnt source parameters $\theta$ is a reasonable approximation to the true optimal $\theta^{\text{opt}}$ on the shifted domain, self-training with the conjugate pseudo-labels provides a reasonable proxy for fine-tuning the network on the true OOD loss. We emphasize that, common to most approaches for TTA, there are still some amount of design decisions that must be put in place; these are detailed in Section 5.1. In practice, we observe OOD generalization typically benefits (across all baselines) from an additional "temperature" scaling, i.e., applying the TTA loss to $h_\theta(x)/T$ for some fixed temperature $T$, although it requires a held-out validation dataset for tuning $T$. However, we should emphasize that truly unsupervised TTA would require making an informed guess for the value of these hyper-parameters. The full procedure for test time adaptation via conjugate pseudo-labels is shown in Algorithm 1.

---

**Algorithm 1** Conjugate pseudo-labeling (Conjugate PL)

---

**Input:** Source classifier $\theta_0$ trained using loss $\mathcal{L}(h_\theta(x), y) = f(h_\theta(x)) - h_\theta(x)^\top y$.
          $N$ batches of test data $D_{\text{test}} = [x_1, x_2, \ldots, x_N]$
**Hyperparams:** learning rate $\eta$ and temperature $T$.

Let $\bar{h}_\theta(x) \stackrel{\text{def}}{=} h_\theta(x)/T$ be the temperature scaled predictor.

Let $\tilde{y}_\theta^{\text{CPL}}(x)$ denote the conjugate pseudo-label function $\tilde{y}_\theta^{\text{CPL}}(x) = \nabla(f(\bar{h}_\theta(x)))$.

**for** $n = 0, 1, \ldots N - 1$ **do**

     $\theta_{n+1} = \theta_n - \eta \nabla \mathcal{L}\left(\bar{h}_\theta(x_n), \tilde{y}_\theta^{\text{CPL}}(x_n)\right)$    [Self-training with conjugate pseudo-labels]

---

# 5   Experiments

In this section, we empirically evaluate the effectiveness and generality of the proposed conjugate pseudo-labeling procedure (Algorithm 1) for test-time adaptation on a variety of datasets.

## 5.1   Setup

**Datasets.**   We evaluate on the three common corruption benchmarks: adapting a classifier trained on CIFAR-10 to CIFAR-10-C, CIFAR-100 to CIFAR-100-C and ImageNet to ImageNet-C [15]. Following the previous works [47, 50], we report the error averaged across corruptions at the highest severity for CIFAR-10/100-C and averaged across corruptions and severity level for ImageNet-C. We also evaluate on three domain adaptation datasets: adapting a classifier trained on SVHN to MNIST, an ImageNet classifier to ImageNet-R [16] and adapting from synthetic to real data in VISDA-C [38].

**Models and Training losses.**   Following previous works on TTA[47, 50], we use ResNet-26 [14] as the source classifier architecture for CIFAR-10/100 experiments, ResNet-18 for SVHN to MNIST and a ResNet-50 for ImageNet and source synthetic data on VisDA-C. We consider source classifiers trained via the following loss functions: the de-facto cross-entropy, recently proposed polyloss [25] and squared loss [18].

**Baselines.**   Our proposed conjugate pseudo-label is the classic approach of self-training with a specific form of pseudo-labels. In self-training, we replace the label $y$ with a pseudo-label $\tilde{y}(x)$ and adapt by optimizing the loss function $\mathcal{L}(h_\theta(x), \tilde{y}(x))$. Note that we could either instantaneously update the pseudo-labels using the current classifier, or generate pseudo-labels once with just the source classifier. Instantaneous updates have been shown to work better for domain adaptation [7, 40], and we perform instantaneous updates for all methods. While we propose using $\tilde{y}^{\text{CPL}}(x) = \nabla f(h_\theta(x))$ (See Section 4.3), we compare to the standard pseudo-labels used in the literature:

- (i) the "hard" pseudo-label (hard PL) where $\tilde{y}(x) = \arg\max_i \big(h_\theta(x)\big)_i$ is the most likely class as predicted by $h_\theta$. As is common in the self-training literature, we perform confidence thresholding.
- (ii) The "soft" pseudo-label (soft PL) where $\tilde{y}(x)$ is obtained by applying a softmax function to the model predictions $h_\theta(x)$.

We also compare with the following recently proposed test-time adaptation methods.

- Entropy Minimization (ENT) [50] minimizes the entropy of model predictions.
- Robust Pseudo-Label [40] where we minimize a robust classification loss, $\mathcal{L}_{\text{rpl}} = q^{-1}(1 - p(i|x)^q)$ where $i = \text{argmax}_j p(j|x)$ and $q \in [0, 1]$.
- MEMO [54] minimizes entropy of a model's outputs across different augmentations of a test input. We implement a *batch version*, where we see multiple test points at once, for fair comparisons.

**TTA methodology.**   Following [50] and [40], we fine-tune by updating the learnable scale and shift parameters of the batch normalization layers across all adaptation losses. For each batch, batch normalization statistics is also updated, as suggested in [41]. We report performance at the end of one round of test-time adaptation over the entire test set.

We tune the learning rate (LR) and temperature (T) on the validation noises in the corruption benchmark by grid-search. LR is selected from $\{1e^{-1}, 1e^{-2}, \ldots 1e^{-4}\}$ and T from $\{1, 2 \ldots 5\}$. All the experiments have been performed on A6000 GPU's. On domain adaptation benchmarks, where there is no held-out target domain, we set T to be $1$ and use the LR suggested by [6, 50]. We use the same hyperparameter tuning protocol across all methods. We single out temperature as a very important hyperparameter, as we discuss in the results below.

## 5.2   Results on classifiers trained with cross-entropy

We study the effectiveness of our proposed conjugate pseudo-labels when the source classifier is trained via cross-entropy loss. In this case, baselines Softmax PL and ENT are the same as Conjugate PL. Thus we omit them in our results. Table 1, reports the performance of various TTA methods.

When the source classifier is trained via cross-entropy, our conjugate pseudo-label algorithm exactly corresponds to entropy minimization with an additional temperature scaling. Entropy minimization as

| Dataset | Temperature (T) | Hard PL | Robust PL | MEMO | Conjugate PL (**ENT**) |
|---------|:---:|---------|-----------|------|------------|
| CIFAR-10-C | ✗ | 13.95 (±0.06) | 13.97 (±0.04) | **12.60** (±0.04) | 13.07 (±0.05) |
|            | ✓ | 13.95 (±0.06) | 12.85 (±0.04) | **12.51** (±0.01) | **12.51** (±0.03) |
| CIFAR-100-C | ✗ | 45.22 (±0.4) | 39.80 (±0.18) | **38.52** (±0.16) | 41.15 (±0.25) |
|             | ✓ | 45.22 (±0.4) | 36.37 (±0.10) | 37.38 (±0.06) | **36.10** (±0.07) |
| ImageNet-C | ✗ | **45.43** (±0.05) | 45.68 (±0.01) | 48.91(±0.03) | 45.82(±0.01) |
|            | ✓ | 45.43 (±0.05) | 45.61 (±0.01) | 48.91(±0.04) | **45.36**(±0.01) |

Table 1: Mean errors when adapting to corruptions using a source classifier trained via cross-entropy loss. Here, conjugate pseudo-labeling becomes softmax-entropy minimization. With the right temperature scaling, softmax-entropy minimization matches or outperforms other approaches. Prior reported gains of other methods over softmax-entropy minimization disappear when we use temperature scaling. For additional context, the source classifier errors without adaptation are: CIFAR-10-C (29.54%), CIFAR-100-C (62.26%), ImageNet-C (61.89%)

proposed in prior work [50] does not tune the temperature parameter, and some newer objectives such as robust PL or MEMO outperform vanilla entropy minimization. For example, on CIFAR-100-C, vanilla ENT obtaines $41.15\%$ average error, while robust PL improves this to $39.80\%$ and MEMO to $38.52\%$. However, with the right temperature scaling, entropy minimization obtains $36.10\%$ error which outperforms the newer objectives (with and without temperature scaling). A similar observation holds for CIFAR-10-C and ImageNet-C as well. Essentially, the gains over vanilla entropy minimization vanish when we do temperature scaling, and entropy minimization (i.e. conjugate pseudo-labeling corresponding to cross-entropy) turns out to be the best objective after all.

### 5.3 Results on classifiers trained with polyloss and squared loss

In the case of cross-entropy, conjugate pseudo-labeling reduces to the familiar notion of entropy minimization. We now explore the performance of our method on different loss functions where the conjugate pseudo-labels differ substantially from entropy minimization (section 4.3). Table 2 presents the results on the corruption benchmarks and Table 3 presents the results on the other domain adaptation datasets for source classifiers trained with PolyLoss.

| Dataset | T | Hard PL | Robust PL | ENT | MEMO | Softmax PL | Conjugate PL (**Ours**) |
|---------|:---:|---------|-----------|-----|------|-----------|-------------|
| CIFAR-10-C | ✗ | 13.81(±0.12) | 14.23(±0.02) | 13.46(±0.06) | 13.23(±0.07) | 14.64(±0.11) | **13.02**(±0.09) |
|            | ✓ | 13.81(±0.12) | 12.45(±0.05) | 12.23(±0.06) | 12.33(±0.04) | 12.26(±0.04) | **12.08**(±0.05) |
| CIFAR-100-C | ✗ | 40.47(±0.05) | 42.86(±0.11) | 40.12(±0.08) | 39.90(±0.05) | 41.00(±0.11) | **38.17**(±0.17) |
|             | ✓ | 40.47(±0.05) | 39.80(±0.08) | 38.23(±0.05) | 39.23(±0.04) | 37.04(±0.06) | **36.83**(±0.08) |
| ImageNet-C | ✗ | 45.44(±0.21) | 46.27(±0.03) | 46.10(±0.03) | 48.21(±0.05) | 44.63(±0.03) | **44.01**(±0.01) |
|            | ✓ | 45.44(±0.21) | 46.27(±0.03) | 45.50(±0.02) | 48.21(±0.04) | 44.45(±0.03) | **44.01**(±0.01) |

Table 2: Mean errors when adapting to corruptions using a source classifier trained via recently proposed Poly-1 Loss [25]. Conjugate pseudo-labeling consistently outperforms all previous approaches. For additional context, source classifier errors without adaptation : CIFAR-10-C (30.22%), CIFAR-100-C (63.91%) and ImageNet-C (62.18%).

First, we note that, across all datasets in Table 2 and Table 3, our conjugate PL approach outperforms all other TTA losses. With polyloss classifiers, entropy minimization is no longer the best method—on CIFAR-100-C, entropy minimization achieves $38.23\%$ error while our conjugate PL achieves $36.83\%$. We see similar *consistent gains* on CIFAR-10-C, ImageNet-C, ImageNet-R and VisDA-C. On digit adaptation tasks from SVHN to MNIST/USPS/MNISTM, where there is a larger shift between source and target, the gains are especially pronounced. Figure 2 compares how the task loss (polyloss $\epsilon = 6$) on the test data decreases as we adapt the model through conjugate PL and other baselines. We use CIFAR-10-C as an example. Observe that our proposed conjugate PL indeed reduces the task loss the most among other baselines.

| Dataset | Source Error | Hard PL | Robust PL | Entropy | Softmax PL | Conjugate PL Ours |
|---------|--------------|---------|-----------|---------|------------|-------------------|
| SVHN → MNIST | 28.33 | 20.21 | 19.73 | 14.28 | 16.54 | **10.73** |
| SVHN → USPS | 31.58 | 23.32 | 26.12 | 23.12 | 24.07 | **21.62** |
| SVHN → MNISTM | 61.69 | 50.73 | 51.35 | 49.33 | 50.47 | **47.59** |
| ImageNet-R | 64.19 | 58.52 | 59.46 | 58.25 | 56.62 | **55.63** |
| VisDA-C | 58.13 | 40.43 | 45.44 | 44.11 | 39.63 | **38.42** |

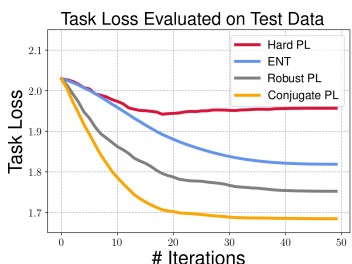

Table 3: Target error when adapting models trained via polyloss on source domains across different domain adaptation benchmarks. Conjugate pseudo-labeling offers consistent and substantial gains over previous approaches across three datasets.

Figure 2: Task Loss (PolyLoss $\epsilon = 6$) evaluated on CIFAR-10-C test data during test-time adaptation.

Furthermore, on CIFAR-10-C and ImageNet-C, we find that adapting polyloss classifiers via conjugate PL improves the performance over all methods applied to cross-entropy trained source classifiers. For e.g., on ImageNet-C, the performance improves from $45.34\%$ to $44.01\%$. However, this is only true when using the proposed conjugate PL. If we just did softmax-entropy minimization (even with temperature scaling), the final adapted performance of a polyloss classifier ($45.5\%$) is in fact worse than that of a cross-entropy classifier ($45.34\%$). Our results suggest that as we develop new training losses that improve the source classifiers, it is important to adapt via conjugate pseudo-labeling to reap the maximum gains.

Similarly, we experiment with the case when the source classifier is trained using squared loss on the CIFAR-10 and CIFAR-100 datasets, and observe consistent gains using the proposed conjugate pseudo-labels over the baselines. For example, on CIFAR-10-C, TTA using conjugate PL gives and error of $12.87\%$, outperforming baselines like ENT ($13.24\%$) and Softmax PL ($31.81\%$). Table 5 in Appendix A.7 shows the detailed results.

Comparing Table 1 and Table 2, we see that the relative ordering between the various baselines differs. This is further evidence that the adaptation loss has to depend on the training loss, and we believe our conjugate pseudo-label approach captures this appropriately by offering consistent gains across the various settings we experimented with.

## 6  Related Works

**Test-time adaptation methods.** In recent years, the setting of test-time adaptation has gained a lot of interest with a host of different approaches proposed in the literature. One family of TTA approaches update the source classifier by minimizing an unsupervised loss on the target distribution [4, 6, 20, 22, 35, 36, 40, 43, 44, 50, 51, 54]. TENT [50] proposes to minimize the entropy of model predictions at test time. Several follow ups like [6, 35, 40, 44, 54] propose alternative TTA objectives, e.g. robust pseudo-labelling [40], likelihood ratio loss [35], entropy of marginal probability averaged across augmentations [54] and self-supervised contrastive losses [6, 49]. However, most of these objectives are heuristically designed or chosen. In this paper, we provide a principled approach of designing unsupervised objectives for TTA .

Another family of approaches for test-time adaptation such as [2, 8, 13, 31, 34, 47] leverage an auxiliary self-supervised task (e.g. rotation prediction [47], masked autoencoders [10]) to update model parameters on each test sample. Crucially, these methods require modifying the source model training by augmenting the supervised training objective with an auxiliary self-supervised loss. Hence it cannot be applied to typical standard classifiers that are trained by minimizing a supervised loss on the source data.

**Source-free domain adaptation.** A very related setting to test-time adaptation is source-free domain adaptation, where a trained source classifier must be adapted to a target distribution of interest, although the entire target unlabeled data is available at once. SHOT [28] proposes to optimize the source hypothesis (i.e. feature extractor) with a combination of entropy minimization, diversity and self-training on pseudo-labels on the unlabeled target data. [53] promotes feature clustering on features from target distributions. [24, 26] use generative modeling to estimate the underlying source distributions for enforcing feature invariance. Such approaches typically require multiple epochs over the target data and cannot be easily adopted to work in an online fashion.

**Unsupervised domain adaptation.** The most canonical setting of domain adaptation involves access to labeled source data and unlabeled target data, all during training. The availability of source and target data during training lends itself to approaches that "align" the source and target representations in some way: [32, 33, 45, 48] match distribution statistics, [11] uses a discriminator, [46] uses self-supervised learning. However, such approaches require access to source data which might not always be feasible due to data privacy and efficiency issues.

**Pseudo-labels and self-training.** Self-training is a classic idea for leveraging unlabeled data, developed first for the semi-supervised setting. Self-training generates pseudo-labels on the unlabeled data, allowing us to use any "supervised" loss on this pseudo-labeled data. Self-training has shown promising results in various settings like semi-supervised learning [19] and improving adversarial robustness [5]. Self-training has also been gaining attention in the setting of unsupervised domain adaptation [28, 39], where pseudo-labels generated on the unlabeled data from target domain is used to supervise the adaptation process. [7, 23, 52] provide theoretical insights into how self-training with pseudo-labels can help under distribution shift. TENT [50] (i.e entropy minimization) can be viewed as a form of self-training with instantaneous softmax pseudo-labels. Our work provides a general framework for the choice of soft pseudo-labels based on the conjugate analysis of the source training objective. Some prior works like [7, 17, 27, 30, 55, 56] have documented the improvement in performance when using instantaneous pseudo-labels over pre-computed pseudo-labels, and thus lend further support to the benefits of our proposed conjugate pseudo-labeling approach. The experiment results presented in this work supporting conjugate pseudo-labels suggest that conjugate pseudo-labels is a promising direction of pseudo-labeling in a broader context.

## 7 Conclusion, Limitations and Future Directions

In this work, we proposed a general test-time adaptation loss, based on the convex conjugate formulation which in turn was motivated by the intriguing meta learning experiments. The fact that meta-learning recovers the proposed loss hints at some kind of optimality of the loss. In Section 4, we prove that for a broad set of loss functions, the proposed (unsupervised) conjugate loss is close to the oracle supervised loss. However, this still does not completely answer what the optimal test-time adaptation loss is and why.

The meta-learning framework in this work was constrained to learn functions over the logits of each individual input. It can be expanded to more involved setups, where we consider functions over the intermediate representations too and also consider learning functions over a batch of input while accounting for their interactions.

Beyond the choice of the adaptation loss itself, achieving good test-time adaptation generally involves several heuristics like updating only the batch norm parameters [50]. While our work was motivated by the loss function, via the meta-learning experiments, we discovered that temperature scaling is another important hyper-parameter that improves the performance of all previous baselines as well. At a high level, test-time adaptation has to be appropriately regularized to prevent the updates over batches from taking the model too far: updating only a few batch norm parameters is one way to do that, and perhaps temperature scaling provides a similar beneficial regularization effect by making the network predictions on unlabeled inputs less confident. Understanding the role of these heuristics more concretely is an interesting direction for future work. It also remains an open problem to understand under what sort of real-world distribution shifts would self-training based approaches would help.

Finally, it is also worth extending and applying the conjugate pseudo-labeling to other settings like semi-supervised learning.

## 8 Acknowledgments

We thank Shubhang Bhatnagar and Asher Trockman for helping with running the ImageNet experiments. We thank Zhili Feng for useful feedback. Sachin Goyal and Mingjie Sun were supported by funding from the Bosch Center for Artificial Intelligence. Aditi Raghunathan was supported by an Open Philanthropy AI Fellowship.

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
