# A Appendix

## A.1 Conjugate Derivations

**Cross-Entropy Loss :**

$$\mathcal{L}(h, y) = -\sum_{i=1}^{c} y_i \log \frac{\exp(h_i)}{\sum_{j=1}^{c} \exp(h_j)}$$
$$= -\sum_{i=1}^{c} y_i * h_i + \log \sum_{j=1}^{c} \exp(h_j) \tag{14}$$
$$= f(h) - y^\top h,$$

where $f(h)$ is $\log \sum_{j=1}^{c} \exp(h_j)$ and the constraint that $\sum_{i=1}^{c} y_i = 1$. Now, the conjugate $f^\star(y)$ is given by :

$$f^\star(y) = -\min_h \{f(h) - y^T h\} = -\min_h \{\log \sum_{j=1}^{c} \exp(h_j) - y^T h\} \tag{15}$$

with the constraint $\sum_{i=1}^{c} y_i = 1$. At the optimality,

$$y_i = (\nabla f(h))_i = \frac{\exp(h_i)}{\sum_j \exp(h_j)} \tag{16}$$

Then,

$$f^\star(y) = -\log \sum_{j=1}^{c} \exp(h_j) + \sum_{i=1}^{c} h_i \frac{\exp(h_i)}{\sum_j \exp(h_j)}$$
$$= \sum_i \frac{\exp(h_i)}{\sum_j \exp(h_j)} \log \frac{\exp(h_i)}{\sum_j \exp(h_j)}, \tag{17}$$

if the constraint $\sum_{i=1}^{c} y_i = 1$ is satisfied, otherwise $f^\star(y) = \infty$ by duality.

This in turn gives, the conjugate loss for cross-entropy (when the constraint is satisfied) :

$$\mathcal{L}^{\text{conj}}(h) = -f^\star(y) = -f^\star(\nabla f(h)) = -\sum_i \frac{\exp(h_i)}{\sum_j \exp(h_j)} \log \frac{\exp(h_i)}{\sum_j \exp(h_j)} \tag{18}$$

**Squared Loss :**

$$\mathcal{L}(h, y) = \frac{1}{2} ||h - y||_2^2$$
$$\approx \frac{1}{2} ||h||_2^2 - y^\top h \text{ [ignoring the constant term]} \tag{19}$$
$$= f(h) - y^\top h,$$

Now, the conjugate $f^\star(y)$ is given by:

$$f^\star(y) = -\min_h \{f(h) - y^T h\} = -\min_h \{\frac{1}{2} ||h||_2^2 - y^T h\}$$
$$= -\frac{1}{2} ||h||_2^2 \tag{20}$$

## A.2 Experiments on Binary Classification with Exponential Loss

Here we present the results on a binary classification task over a synthetic dataset of 100 dimensional gaussian clusters.

**Dataset Creation** For the binary classification task, we create a synthetic dataset similar to [23]. Specifically, let the data $\mathcal{X} \sim \mathcal{N}(\mu, \Sigma) \in \mathbb{R}^{100}$ and labels $\mathcal{Y} \in \{-1, +1\}$. We sample $\mu \sim \mathcal{N}(k, \mathcal{I}_{100})$. For $\Sigma$, similar to [23], we sample a diagonal matrix $D$, where each entry is sampled uniformly from a specified range, and a rotation matrix $U$ from a HAAR distribution, giving $\Sigma = UDU^T$.

For the source data, we sample $\mu_s^{-1}, \mu_s^{+1}, \Sigma_s^{-1}, \Sigma_s^{+1}$ as specified above with $k = 0$. Now to create a distribution shifted data of various severity, we sample $\mu_t^{-1}, \mu_t^{+1}, \Sigma_t^{-1}, \Sigma_t^{+1}$ as specified above with $k = 1$, which are then used to sample the shifted data as follows :

$$\mu_\lambda^1 = \lambda\mu_t^1 + (1 - \lambda)\mu_s^1$$

$$\mu_\lambda^{-1} = \lambda\mu_t^{-1} + (1 - \lambda)\mu_s^{-1}$$

$$\Sigma_\lambda^1 = \lambda\Sigma_t^1 + (1 - \lambda)\Sigma_s^1$$

$$\Sigma_\lambda^{-1} = \lambda\Sigma_t^{-1} + (1 - \lambda)\Sigma_s^{-1}$$

$$\mathcal{X}_\lambda \sim \mathcal{N}(\mu_\lambda, \Sigma_\lambda)$$

In the following experiments, easy shift refers to $\lambda = 0.6$, moderate shift to $\lambda = 0.65$ and hard shift to $\lambda = 0.7$.

**Exponential Loss for Binary Classification** Let $z$ be the classification score $h_\theta(x)$. For logistic training loss, conjugate adaptation loss would default to entropy with sigmoid probability. Thus, here we experiment with a different but also commonly used surrogate loss to 0/1 loss: exponential loss, which is defined as:

$$\mathcal{L}_{\exp}(z, y) = \exp(-yz) \tag{21}$$

where $y \in \{-1, +1\}$. It can be rewritten in the expanded conjugate form of:

$$\mathcal{L}_{\exp}(z, y) = \frac{1}{2} \cdot \left(e^z + e^{-z}\right) - \frac{1}{2} \cdot y \cdot \left(e^z - e^{-z}\right) \tag{22}$$

For exponential loss, the conjugate pseudo-label function and the conjugate pseudo-label loss are:

$$y_{\exp}^{\text{CPL}}(z) = \frac{e^z - e^{-z}}{e^z + e^{-z}}, \quad \mathcal{L}_{\exp}^{\text{CPL}}(z) = \frac{2}{e^z + e^{-z}} \tag{23}$$

The model is adapted on shifted gaussian clusters and we compare the conjugate loss with two baseline approaches: 1) Hard pseudo-labelling $\exp(-y_{\text{hard pl}} \cdot z)$; 2) Entropy applied to sigmoid probability $P(y = +1) = \sigma(z)$. The losses are compared on three degrees of shift (easy, moderate and hard), which is controlled by the drifted distance of Gaussian clusters. The results are shown in Figure 3, where we plot the accuracy curve with respect to adaptation iterations. With easy and moderate shift, conjugate loss (green) generalizes faster to shifted test data; with hard shift, only conjugate loss improves model accuracy on shifted test data while entropy (blue) deteriorates model performance.

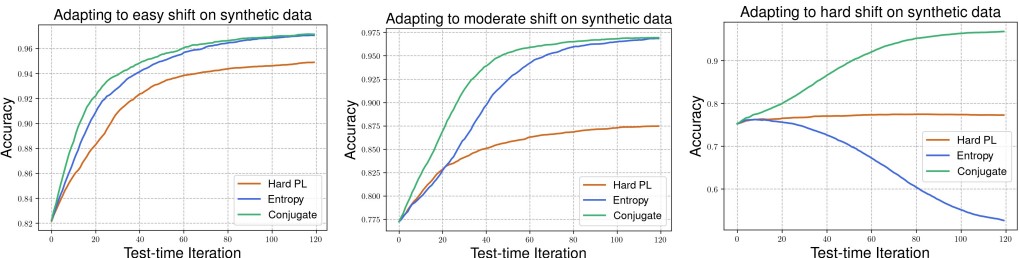

Figure 3: Test-time adaptation result on synthetic data with three shift levels ranging from easy, moderate and hard (detailed in section A.2). The source model is a linear classifier trained with exponential loss $\mathcal{L}_{\exp} = e^{-yh_\theta(x)}$. Adaptation with the conjugate loss generalizes better compared to baseline losses.

## A.3  Meta Learning Experiment Details

In section 3 we talked about learning the meta-loss function parameterized by a neural network $m_\phi : \mathbb{R}^{|\mathcal{Y}|} \mapsto \mathbb{R}$, that takes in the model predictions/logits and outputs a loss value. Here we discuss the architecture chosen and the implementation details. Further, in Appendix A.4 we empirically show that the learnt meta-loss is not affected by the choice of task loss / surrogate loss used in meta learning ($\mathcal{L}$ in Equation 1). Note that the task loss / surrogate loss function is used to update the meta-loss $m_\phi$ during meta-learning. The surrogate loss is calculated on updated source model's predictions on labeled samples from test domain. The surrogate loss tries to update the meta-loss in the outer loop such that when meta-loss is later used to update the source model in the inner loop, the source model generalizes better to the test domain.

**Architecture and Implementation Details**  Figure 4 gives an overall schema for meta-learning the loss function and algorithm 2 gives the pseudo-code for meta-learning the loss function. Below we describe this in further detail. We use a transformer (denoted by $\mathcal{T}$) with a MLP (denoted by $\mathcal{P}$) over the output of transformer as the architecture for $m_\phi$, i.e. $m_\phi(x) = \mathcal{P}(\mathcal{T}(x))$. Specifically, for a given source trained model $h_\theta$ and input $x \sim D_{\text{test}}$ :

1. Let $h_\theta(x) \in \mathbb{R}^{|\mathcal{Y}|}$ be the model predictions/logits, where $|\mathcal{Y}|$ denotes the number of classes.

2. Let $h_\theta^j(x) \in \mathbb{R}, \forall j \in |\mathcal{Y}|$ be the prediction corresponding to class $j$.

3. The input to transformer is then given by $z \in \mathbb{R}^{|\mathcal{Y}| \times (1+e)}$, where $z^j \in \mathbb{R}^{1+e}, \forall j \in |\mathcal{Y}|$ is the concatenation of $h_\theta^j(x)$ and the learnable positional embedding $pe_j \in \mathbb{R}^e$.

4. The transformer output is given by $w = \mathcal{T}(z) \in \mathbb{R}^d$, where $d$ denotes the feed-forward dimension of the transformer.

5. The transformer output $w$ is finally passed through a MLP to get the meta-loss value $m_\phi(h_\theta(x)) = \mathcal{P}(w) \in \mathbb{R}$

6. The source model is updated by optimizing over the meta-loss.

$$\theta^{t+1} \leftarrow \theta^t - \alpha \frac{\partial m_{\phi^t}(h_{\theta^t}(x))}{\partial \theta^t} \tag{24}$$

7. The updated source model is then used to update the meta-loss by optimizing over some supervised loss function $\mathcal{L}_{\text{task}}$.

$$\phi^{t+1} \leftarrow \phi^t - \beta \frac{\partial \mathcal{L}_{\text{task}}(h_{\theta^{t+1}}(x'), y')}{\partial \phi^t}, \quad \text{where } (x', y') \sim D_{\text{test}} \tag{25}$$

Note that the last step assumes access to labels of test inputs. In this paper, we do not propose meta-learning the TTA loss as an approach. Rather, we use meta-learning to explore what the "best" TTA losses look like.

We select the trasformer input embedding dimension $(1 + e)$ from $\{16, 32, 64\}$ and transformer feed-forward dimension $d$ from $\{32, 64, 128\}$. The number of transformer layers and the hidden layers in MLP are selected from $\{1, 2\}$. We use Adam optimizer with a learning rate of $1e^{-3}$ for learning the meta-loss (i.e. the transformer + MLP). We train the meta-loss for 100 epochs with a batch size of 200.

## A.4  Effect of Task Loss in Meta Learning

In section 3, we show that the meta losses learned on different source classifiers differ substantially if the source classifiers are trained using different source loss functions. Here we further empirically verify that the learnt meta loss is not affected by the task loss used in meta learning ($\mathcal{L}$ in Equation 1). Thus the learnt meta loss is determined by the source model.

In Figure 5, we show the meta loss learnt on a ResNet-26 trained with Cross Entropy loss for two meta task losses: Cross Entropy Figure 5a and Squared Loss Figure 5b. We plot the meta loss as a function over one of its input prediction scores, while keeping other fixed. We can see that the task loss barely affects the learnt meta loss. Similar observations can be made for the classifier trained with squared loss Figure 6.

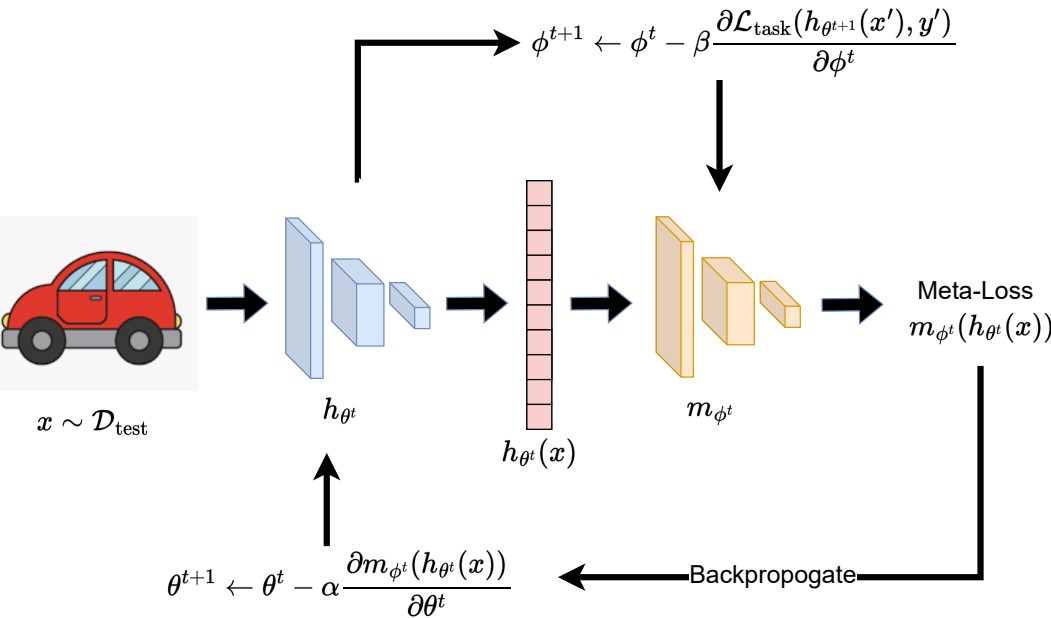

Figure 4: Meta-Loss learning procedure : The model predictions $h_{\theta^t}(x)$ are passed through the parameterized loss function $m_{\phi^t}$, which outputs a loss value. We optimize $\phi$ such that when optimizing the source model over the loss $m_{\phi^t}(h_{\theta^t}(x))$, the updated $\theta^{t+1}$ has a better performance on the test domain. To do this, we take one gradient step over the meta-loss to get the update source model parameters $\theta^{t+1}$, and then update $\phi$ by evaluating $\theta^{t+1}$ on the labeled validation data using some task loss $\mathcal{L}_{\text{task}}$.

---

**Algorithm 2** Learning the Meta-Loss

**Input:** Source trained classifier $h_{\theta^0}$. Randomly initialized meta-loss $m_{\phi^0}$.
   Task loss / Surrogate loss $\mathcal{L}_{\text{task}}$ like cross-entropy or squared loss for meta learning
   $N$ batches of test data $D_{\text{test}} = [(x_1, y_1), \ldots, (x_N, y_N)]$
**Hyperparams:** learning rates $\alpha$ and $\beta$.

   **for** $epoch = 0, 1, 2, \ldots$ **do**
     **for** $n = 0, 1, \ldots N - 1$ **do**
       $\theta^{t+1} \leftarrow \theta^t - \alpha \frac{\partial m_{\phi^t}(h_{\theta^t}(x_n))}{\partial \theta^t}$
       Sample $(x_r, y_r) \sim D_{\text{test}}$.
       $\phi^{t+1} \leftarrow \phi^t - \beta \frac{\partial \mathcal{L}_{\text{task}}(h_{\theta^{t+1}}(x_r), y_r)}{\partial \phi^t}$

---

## A.5   Test-Time Adaptation Detail

For completeness, we also give the test-time adaptation setup in Algorithm 3.

## A.6   ImageNet results on each severity level

In continuation with results shown in Table 2 in Section 5.3, Table 4 shows the mean errors averaged across the 15 corruption types for each of the severity level on ImageNet-C, for a source classifier trained with PolyLoss ($\epsilon = 8$).

## A.7   Square Loss Trained Source Classifier

In Section 5.3, we briefly discussed that similar to the other source training losses like cross-entropy and polyloss, our proposed conjugate loss outperforms the baselines when the source classifier is

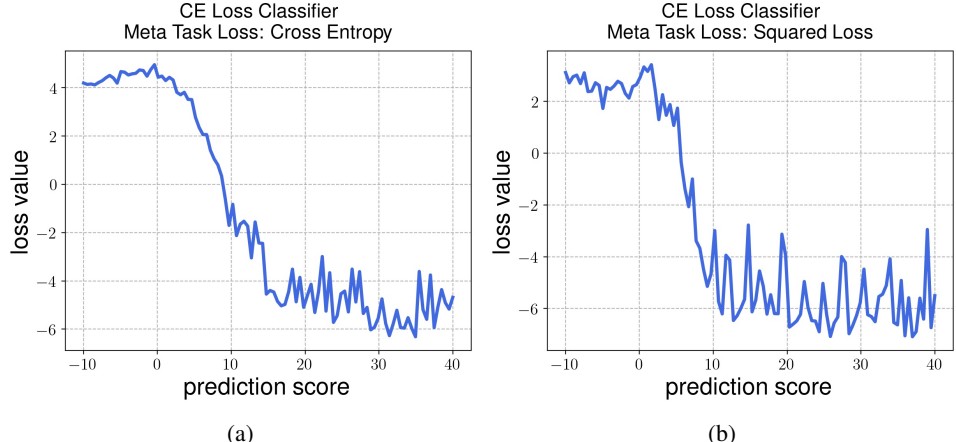

Figure 5: Visualizations of meta loss by varying one input dimension (prediction score). The source model is a ResNet-26 trained with Cross Entropy. Here we show meta loss trained by two different task losses: Cross Entropy Figure 5a and Squared Loss Figure 5b.

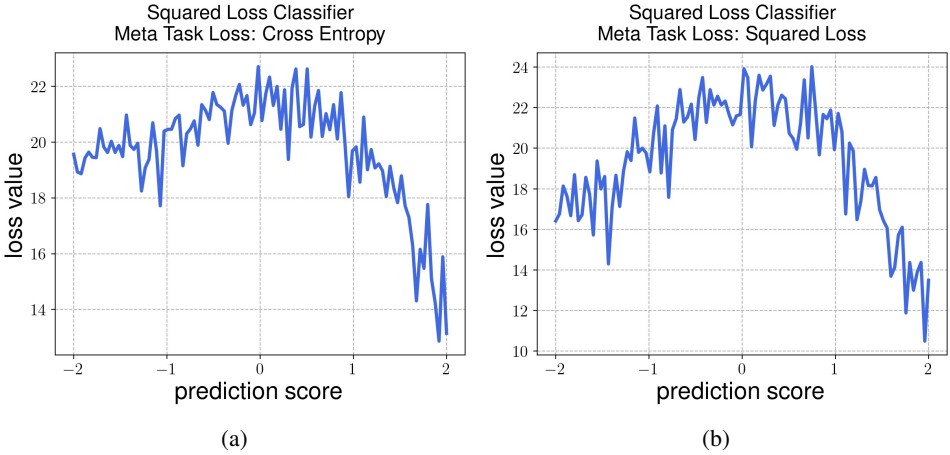

Figure 6: Visualizations of meta loss by varying one input dimension (prediction score). The source model is a ResNet-26 trained with Squared Loss. Here we show meta loss trained by two different task losses: Cross Entropy Figure 6a and Squared Loss Figure 6b.

---

**Algorithm 3** Test-Time Adaptation

---

**Input:** Source classifier $\theta_0$ trained using loss $\mathcal{L}(h_\theta(x), y)$, An unsupervised loss function for test-time adaptation $\mathcal{L}_{\text{tta}}(x)$, $N$ batches of test data $D_{\text{test}} = [x_1, \ldots, x_N]$
**Hyperparams:** learning rate $\eta$.

**for** $n = 0, 1, \ldots N - 1$ **do**
    $\theta_{n+1} = \theta_n - \eta \nabla \mathcal{L}_{\text{tta}}(x_n)$

    $\hat{y}_n = h_{\theta_{n+1}}(x_n)$  [Predictions for the $n^{th}$ batch]

---

| Corrution Severity | Temperature | Robust PL | Entropy | MEMO | Softmax PL | Conjugate |
|---|---|---|---|---|---|---|
| 1 | ✗ | 34.27 | 33.17 | 34.39 | 32.49 | **32.26** |
| | ✓ | 34.27 | 32.84 | 34.39 | 32.70 | **32.26** |
| 2 | ✗ | 41.25 | 39.04 | 40.38 | 37.78 | **37.40** |
| | ✓ | 41.25 | 38.50 | 40.38 | 37.75 | **37.40** |
| 3 | ✗ | 47.37 | 44.04 | 45.67 | 42.30 | **41.72** |
| | ✓ | 47.37 | 43.33 | 45.67 | 42.14 | **41.72** |
| 4 | ✗ | 56.63 | 51.88 | 54.49 | 49.61 | **48.84** |
| | ✓ | 56.63 | 51.03 | 54.49 | 49.39 | **48.84** |
| 5 | ✗ | 67.11 | 62.53 | 66.13 | 60.94 | **59.90** |
| | ✓ | 67.11 | 61.80 | 66.13 | 60.30 | **59.90** |
| Mean | ✗ | 49.32 | 46.13 | 48.21 | 44.62 | **44.02** |
| | ✓ | 49.32 | 45.50 | 48.21 | 44.45 | **44.02** |

Table 4: Mean Errors across the 15 noises for various severity level on the ImageNet-C dataset, with source model trained using Poly-1 Loss. Note that Temperature scaling helped only in the case of Entropy and Softmax PL.

trained using a squared loss. Table 5 shows a detailed comparison with the baselines. We note that for the conjugate of squared loss, the temperature scaling can be wrapped into the learning rate as shown in Section 4.2. Further, on the CIFAR-10-C dataset we observe temperature scaling doesn't help any of the other baselines too, hence we do not include the temperature row in CIFAR-10-C.

| Dataset | Temperature | Hard PL | Robust PL | ENT | MEMO | Softmax PL | Conjugate PL |
|---|---|---|---|---|---|---|---|
| CIFAR-10-C | ✗ | 13.71 ($\pm$0.07) | 13.06 ($\pm$0.05) | 13.24 ($\pm$0.02) | 13.22 ($\pm$0.04) | 14.85 ($\pm$0.08) | **12.99** ($\pm$0.04) |
| CIFAR-100-C | ✗ | 50.82 ($\pm$0.31) | 44.53 ($\pm$0.13) | 43.55 ($\pm$0.12) | 51.35 ($\pm$0.04) | 51.99 ($\pm$0.03) | **43.39** ($\pm$0.11) |
| | ✓ | 50.82 ($\pm$0.31) | 43.99 ($\pm$0.15) | **43.21** ($\pm$0.08) | 51.35 ($\pm$0.04) | 51.99 ($\pm$0.03) | 43.39 ($\pm$0.11) |

Table 5: Mean Errors on the common corruptions datasets for source classifier trained using squared loss. We note that temperature scaling didn't help on the CIFAR-10-C dataset. Source Classifier Errors without adaptation : CIFAR-10-C (28.34%), CIFAR-100-C (68.79%)

| Dataset | Temperature (T) | Hard PL | Robust PL | MEMO | Conjugate PL (**ENT**) |
|---|---|---|---|---|---|
| CIFAR-10-C | ✗ | SGD,$1e^{-3}$, 1 | SGD,$1e^{-3}$, 1 | SGD,$1e^{-3}$, 1 | SGD, $1e^{-3}$, 1 |
| | ✓ | SGD,$1e^{-3}$, 1 | SGD,$1e^{-2}$, 2 | SGD,$5e^{-3}$, 3 | Adam,$1e^{-3}$, 2 |
| CIFAR-100-C | ✗ | SGD,$1e^{-2}$, 1 | SGD,$1e^{-2}$, 1 | SGD,$5e^{-3}$, 1 | SGD, $1e^{-2}$, 1 |
| | ✓ | SGD,$1e^{-2}$, 1 | SGD,$1e^{-2}$, 2 | SGD,$1e^{-2}$, 2 | SGD,$1e^{-2}$, 2 |
| ImageNet-C | ✗ | SGD,$1e^{-2}$, 1 | SGD,$2.5e^{-3}$, 1 | SGD,$1e^{-3}$, 1 | SGD,$2.5e^{-3}$, 1 |
| | ✓ | SGD,$1e^{-2}$, 1 | SGD,$2.5e^{-3}$, 1.5 | SGD,$1e^{-3}$, 1 | SGD,$2.5e^{-3}$, 1.5 |

Table 6: Hyper-parameters (**Optimizer, Learning Rate, Temperature**) for the results in Table 1, where we showed the mean errors on the common corruptions dataset for a source classifier trained using cross-entropy loss.

## A.8 Hyper-Parameters

We share the exact hyper-parameters found using gridsearch over the 4 validation noises for the common corruptions dataset.

**Cross Entropy Classifier Experiments**  In Section 5.2, Table 1 shows the results when adapting a cross entropy trained classifier on various common corruptions dataset. Table 6 gives the optimizer, learning rate and optimal temperature for each of the baseline and our proposed conjugate loss.

**PolyLoss Classifier Experiments**  In Section 5.3, Table 2 shows the results when adapting a polyloss trained classifier on various common corruptions dataset. Table 7 gives the optimizer, learning rate and optimal temperature for each of the baseline and our proposed conjugate loss.

| Dataset | T | Hard PL | Robust PL | ENT | MEMO | Softmax PL | Conjugate PL (**Ours**) |
|---|---|---|---|---|---|---|---|
| CIFAR-10-C | ✗ | SGD,$1e^{-3}$, 1 | SGD,$1e^{-3}$, 1 | SGD,$1e^{-3}$, 1 | SGD,$5e^{-3}$, 1 | SGD, $1e^{-3}$, 1 | SGD, $1e^{-3}$, 1 |
|  | ✓ | SGD,$1e^{-3}$, 1 | SGD,$1e^{-2}$, 3 | SGD,$1e^{-2}$, 3 | SGD,$5e^{-3}$, 3 | SGD, $1e^{-3}$, 2 | SGD, $1e^{-3}$, 1.5 |
| CIFAR-100-C | ✗ | SGD,$1e^{-2}$, 1 | SGD,$1e^{-2}$, 1 | SGD,$1e^{-2}$, 1 | SGD,$1e^{-2}$, 1 | SGD, $1e^{-2}$, 1 | SGD, $1e^{-2}$, 1 |
|  | ✓ | SGD,$1e^{-2}$, 1 | Adam,$1e^{-3}$, 3 | SGD,$1e^{-2}$, 2 | SGD,$1e^{-2}$, 2 | SGD, $1e^{-2}$, 2.5 | SGD, $1e^{-2}$, 1.5 |
| ImageNet-C | ✗ | SGD,$1e^{-2}$, 1 | SGD,$2.5e^{-3}$, 1 | SGD,$2.5e^{-3}$, 1 | SGD,$5e^{-3}$, 1 | SGD, $2.5e^{-3}$, 1 | SGD, $2.5e^{-3}$, 1 |
|  | ✓ | SGD,$1e^{-2}$, 1 | SGD,$2.5e^{-3}$, 1 | SGD,$2.5e^{-3}$, 1.5 | SGD,$5e^{-3}$, 1 | SGD, $2.5e^{-3}$, 2 | SGD, $2.5e^{-3}$, 1 |

Table 7: Hyper-parameters (**Optimizer, Learning Rate, Temperature**) for the results in Table 2, where we showed the mean errors on the common corruptions dataset for a source classifier trained using poly-loss.

**Squared Loss Classifier Experiments**  In Section 5.3, we briefly discussed the results when adapting a squared loss trained classifier on various common corruptions dataset. Table 8 gives the optimizer, learning rate and optimal temperature for each of the baseline and our proposed conjugate loss for the results in Table 5.

**Digit Adaptation Datasets**  For the experiments on digits adaptation tasks, we do not have any validation set. Hence, we don't use temperature scaling here ($T = 1$) and fix the optimizer and LR as Adam and $1e^{-2}$ respectively for all the baselines.

### A.9   Additional Experiments on Digit Adaptation Datasets

Similar to the setting of Table 1, we perform additional experiments on digit adaptation datasets when the source classifier is trained using the cross-entropy loss. Note that when the source classifier is trained using cross-entropy loss, the conjugate loss is equal to the softmax-entropy. In the absence of validation dataset in digit adaptation benchmarks, we used a fixed learning rate of 0.01 for all the baselines, optimizer as Adam and an informed temperature scaling guess of T=2.

Table 9 compares softmax-entropy minimization with various baselines. Here, again we observe that on SVHN → MNIST benchmark, without temperature scaling, MEMO (10.67% error) outperforms softmax-entropy (14.41% error). However, similar to the observations in Table 1, with temperature scaling, softmax-entropy minimization (9.26% error) is able to match the performance of MEMO (9.36% error). Further, on the SVHN → USPS benchmark, softmax-entropy (conjugate) and MEMO perform similar even without temperature scaling.

### A.10   Additional Meta Learning the TTA Loss Experiments

In Section 3, we tried to learn a test-time adaptation (TTA) loss via meta-learning for adapting a CIFAR10 trained ResNet26 to distribution shifts on CIFAR10 corruptions. Figure 1 showed that the learnt meta-loss looks like a temperature scaled softmax-entropy.

In this section, we show the learnt meta loss across a range of settings as described below :

1. Digit Adaptation: Figure 7a and 7b show the learnt meta-loss when adapting a SVHN trained ResNet26 to MNIST dataset and USPS dataset respectively. We observe that the learnt meta-loss can be well approximated by a temperature scaled softmax-entropy.

2. Various Noise Types: In Figure 8, we show the learnt meta-loss when adapting a ResNet26 trained on CIFAR10 dataset using cross-entropy loss, to various noise types like speckle, gaussian, saturate and spatter. The severity level is kept fixed at the maximum i.e. 5.

| Dataset | T | Hard PL | Robust PL | ENT | MEMO | Softmax PL | Conjugate PL (**Ours**) |
|---|---|---|---|---|---|---|---|
| CIFAR-10-C | ✗ | SGD,$1e^{-2}$, 1 | SGD,$1e^{-2}$, 1 | SGD,$1e^{-2}$, 1 | SGD,$1e^{-2}$, 1 | SGD,$1e^{-4}$, 1 | SGD,$1e^{-2}$, 1 |
| CIFAR-100-C | ✗ | Adam,$1e^{-3}$, 1 | Adam,$1e^{-3}$, 1 | Adam,$1e^{-3}$, 1 | Adam,$1e^{-3}$, 1 | Adam, $1e^{-4}$, 1 | Adam, $1e^{-3}$, 1 |
|  | ✓ | Adam,$1e^{-3}$, 1 | Adam,$1e^{-3}$, 0.5 | Adam,$1e^{-3}$, 2 | Adam,$1e^{-3}$, 2 | Adam, $1e^{-4}$, 2.5 | Adam, $1e^{-3}$, 1 |

Table 8: Hyper-parameters (**Optimizer, Learning Rate, Temperature**) for the results in Table 5, where we showed the mean errors on the common corruptions dataset for a source classifier trained using squared loss.

| Dataset | Temperature (T) | Hard PL | Robust PL | MEMO | Conjugate PL (**ENT**) |
|---|---|---|---|---|---|
| SVHN → MNIST | ✗ | 21.54 | 27.44 | **10.67** | 14.41 |
|  | ✓ | 21.54 | 13.26 | **9.36** | **9.26** |
| SVHN → USPS | ✗ | 26.06 | 26.81 | 22.72 | **22.57** |
|  | ✓ | 26.06 | **22.32** | 22.42 | **22.27** |

Table 9: Mean errors when adapting to digit adaptation benchmarks using a source classifier trained via cross-entropy loss. Here, conjugate pseudo-labeling becomes softmax-entropy minimization. Again we observe that with the right temperature scaling, softmax-entropy minimization matches other approaches. For additional context, the source classifier errors without adaptation are: SVHN → MNIST (34.17%), SVHN → USPS (31.84%).

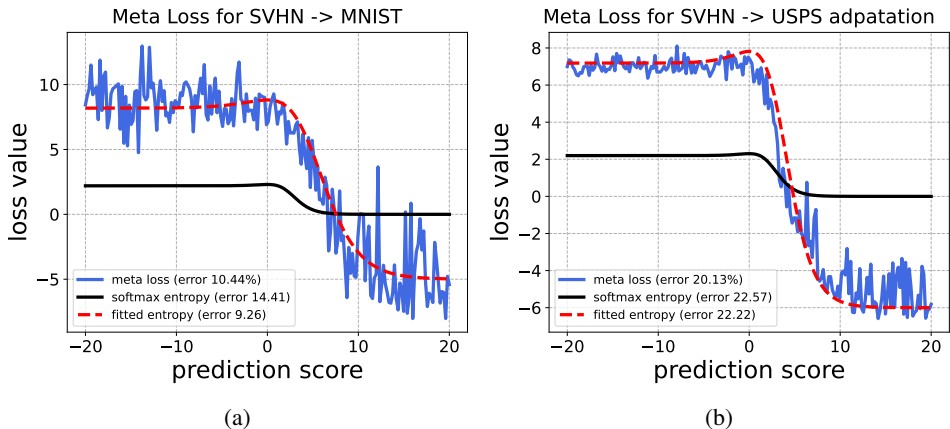

(a)                 (b)

Figure 7: Visualizations of the learnt meta-loss by varying one input dimension (prediction score). The source model is a ResNet-26 trained with cross-entropy on the SVHN dataset. (a) The learnt meta-loss when adapting to the MNIST test dataset. (b) The learnt meta-loss when adapting to the USPS test dataset.

3. Various Severity Levels: In Figure 9, we vary the severity level of the noise, keeping the noise type fixed.

4. Dataset and Architecture: In Figure 10, we compare the learnt meta-loss when adapting to speckle noise, for different source classifier architectures (ResNet26 and ResNet50) and different source training dataset (CIFAR10 and CIFAR100). In all the cases, we again observe that the learnt meta-loss can be well approximated by a temperature scaled softmax-entropy.

5. Squared Loss : Finally, in Figure 11 we show the learnt meta-loss for classifiers trained with squared loss function instead of cross-entropy. We observe that in this case, the learnt meta loss mimics a quadratic function as expected from the conjugate formulation.

For each of the learnt meta losses, we also show the values $(\alpha, T, C)$ we use to fit the meta loss with softmax entropy function: $\alpha \cdot \mathcal{H}(\mathrm{softmax}(x/T)) - C$. Note that although the learnt meta-loss can be approximated by the conjugate, the parameters $\alpha, T, C$ differ across the settings.

In the case of classifiers trained with squared loss, we fit the meta loss with a quadratic function $\sum_{i=1}^{K}(A \cdot x_i^2 + C)$, where $K$ is the number of classes and $x$ is the logit vector. Again, we also show the fitted parameter value $A, C$. The meta loss follows the trend of a quadratic function. The fitted quadratic function performs better or similar as the meta loss, while the parameters of the fitted quadratic function remain different across the meta learning setup (base classifier architectures and noise types).

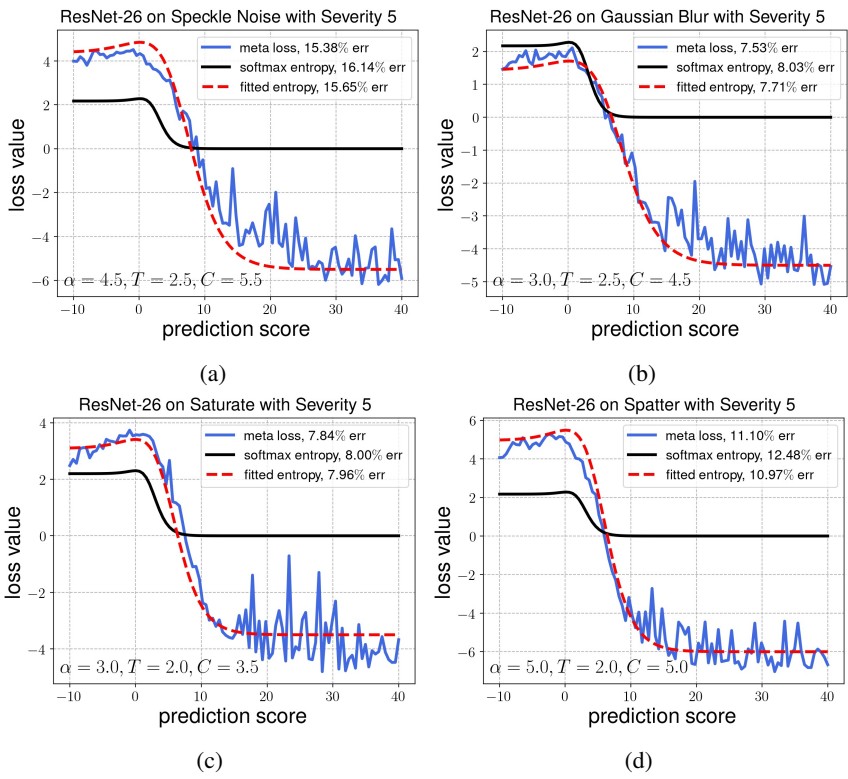

Figure 8: Visualization of meta loss (blue) learnt from various noise types in CIFAR-10-C validation set, where base classifiers are trained with cross-entropy loss. We show the error of meta loss, softmax entropy and fitted entropy for test-time adaptation on the corresponding noise types. We also show the parameters $(\alpha, T, C)$ in the fitted entropy.

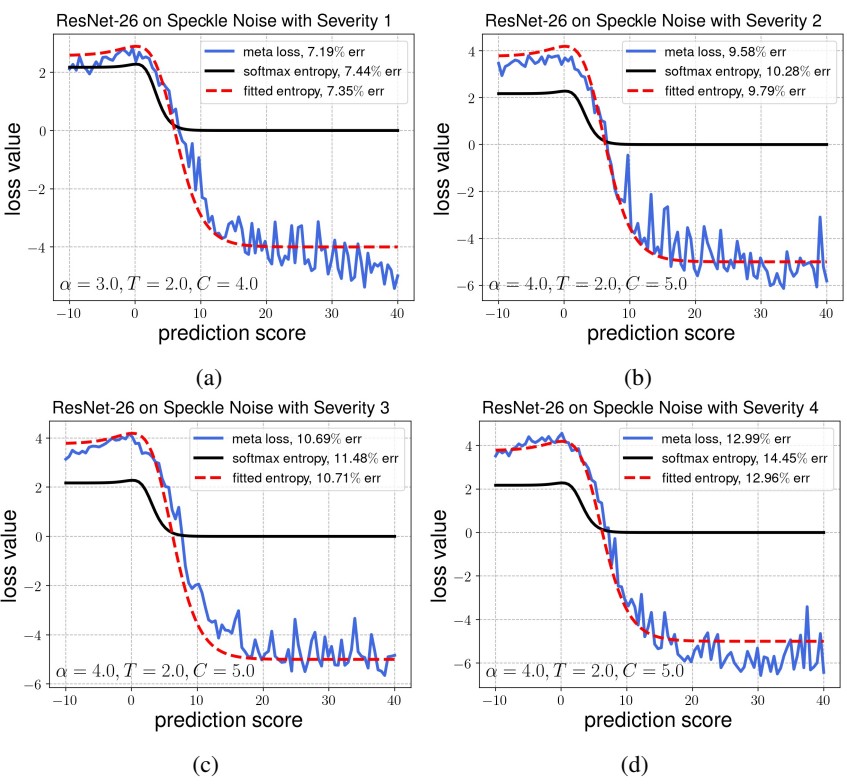

Figure 9: Visualization of meta loss (blue) learnt on speckle noise with different severity level for CIFAR-10-C, where base classifiers are trained with cross-entropy loss. We show the error of meta loss, softmax entropy and fitted entropy for test-time adaptation on the corresponding noise types. We also show the parameters $(\alpha, T, C)$ in the fitted entropy.

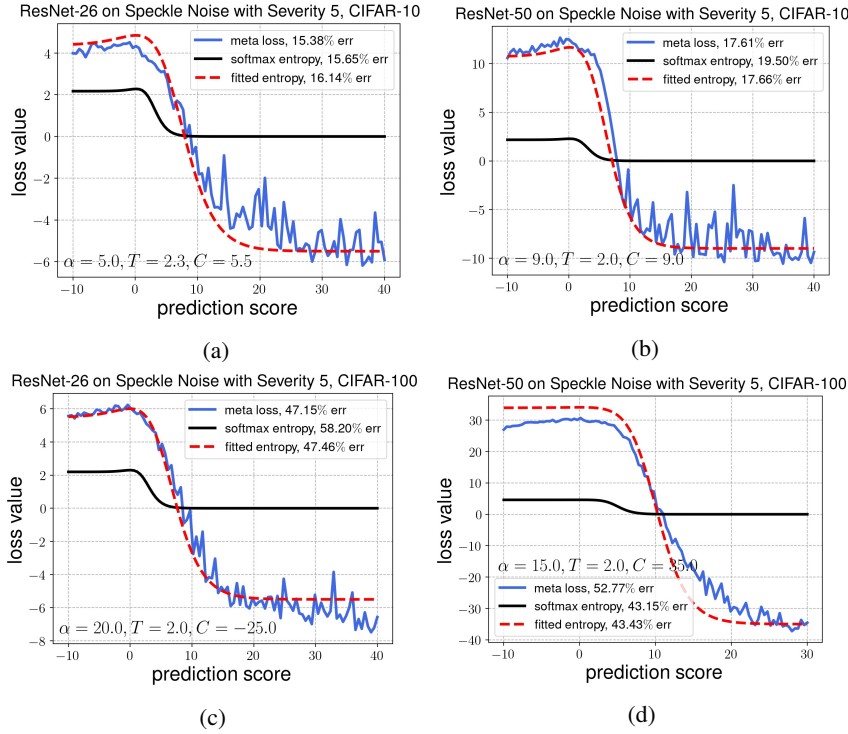

(a)

(b)

(c)

(d)

Figure 10: Visualization of meta loss (blue) learnt across datasets (CIFAR-10-C/CIFAR-100-C) and base classifier architectures (ResNet-26/ResNet-50), where base classifiers are trained with cross-entropy loss. We show the error of meta loss, softmax entropy and fitted entropy for test-time adaptation on the corresponding noise types. We also show the parameters $(\alpha, T, C)$ in the fitted entropy.

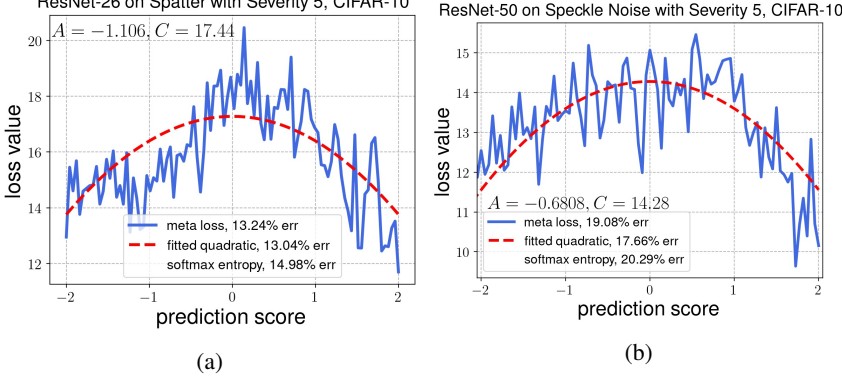

(a)

(b)

Figure 11: Visualization of meta loss (blue), where base classifier is trained with quadratic loss. We show the error of meta loss, softmax entropy and fitted quadratic function for test-time adaptation on the corresponding noise types. We also show the parameters $(A, B, C)$ in the fitted quadratic function.