# OpenReview forum: "Test Time Adaptation via Conjugate Pseudo-labels"
_NeurIPS.cc/2022/Conference — NeurIPS 2022 Accept_

### Official Review · Reviewer_U1Uv · 2022-07-11

**Rating:** 7
**Confidence:** 4
**Soundness:** 3 good
**Presentation:** 4 excellent
**Contribution:** 3 good

**Summary:**

Paper proposes a general way of obtaining test-time adaptation loss which is used for more robust predictions under distribution shifts. First, this loss is learnt using meta-learning and the insights obtained from the same is used to obtain the proposed TTA loss (related to the conjugate of the training loss). The proposed loss is then reinterpreted as self-training with pseudolabels.

**Questions:**

1. I do not agree with the paper’s claims about the “best” TTA loss as:
- “Best” loss is not defined theoretically; Equation (6) is an approximation and it is unclear how close an approximation it is.
- The scale of the “best” TTA loss from meta-learning seems to vary with task loss (Figure 5 & 6) which suggests that Equation (6) that has no scale is not the “best”. (My concern with temperature scaled TTA loss is below; point 2).
- “Best” TTA loss likely depends on the type of distribution shifts. The meta-learning experiments are done on CIFAR10 with noises/corruptions. But the theory does not discuss what kind of distribution shifts are allowed. For example, is the only requirement that learnt $\theta^\star$ is (close to) optimal for target domain as well? Could we observe different meta-learnt loss if the distribution shifts were stronger, say shifts in illumination etc.?

2. My other main concern is regarding the temperature scaling.

- The paper does not provide a reasonable theoretical justification of using a temperature parameter since the approximation in Equation (6) doesn’t seem to demand it. It seems to be needed empirically though, as seen from Figure 1a and results in Section 5, but it is not clear why.

- The need for temperature scaling creates an issue that held-out labeled validation data from target domain is required to tune it. Further, the paper also uses validation target domain data to tune the learning rate as well. This setting is easier than the standard TTA setting and many other domain adaptation methods could be more applicable. For example, I believe the relevant prior work TENT [1] does not assume the presence of this validation data and does no tuning.


- While the paper claims that conjugate PL is much less sensitive to tuning T, I am not convinced it is. Table 1 shows that proposed approach + cross-entropy training loss is never best without temperature scaling and becomes best with temperature scaling. It would be helpful to see if similar trends appear on other domain adaptation datasets like SVHN->MNIST, etc. with the cross-entropy loss. In Table 2, while conjugate PL outperforms other loss functions, the sensitivity to temperature scaling is still comparable with other methods (compare ENT or MEMO).

3. Minor:
- More detail regarding Equation 1 will help the reader unfamiliar with meta learning loss functions. For example, the role of $\mathcal{L}$, clarifying its difference from the source training loss.

- Section 4.1 should include a reference for the optimization problem and its solution.

- $h^\star$ is not defined; I assume it is the optimal function. It is better to use another symbol to avoid confusion with conjugate symbol. At several places (like equations 8 and 9), $f^\star$ is used instead of $f^*$.

**AFTER REBUTTAL**: The authors have adequately addressed my concerns and updated the paper. I have increased my score.


References:

[1] Dequan Wang, Evan Shelhamer, Shaoteng Liu, Bruno Olshausen, and Trevor Darrell. Tent: Fully test-time adaptation by entropy minimization. In International Conference on Learning Representations, 2021.



**Limitations:**

It would be helpful to clarify the following limitations better: What kind of distribution shifts are allowed? When is the proposed loss "best" with/without temperature scaling?

**Strengths And Weaknesses:**

**Strengths:**

1. Paper is novel and useful as it provides a general framework to perform test-time adaptation for any training loss. Further, it provides an informal theory on why existing TTA loss (softmax-entropy) is a good TTA loss.

2. Paper is clearly written and well-motivated through preliminary experiments.

3. Experiments show advantage of the proposed training procedure, particularly in more recent non-standard loss functions.

**Summary of weaknesses (details provided later):**

1. The paper’s claims about the proposed loss being “best” TTA loss are not substantiated theoretically.

2. The use of the temperature parameter is not motivated theoretically; it seems like an empirical fix.

3. Tuning temperature parameter requires validation data from some target domains which is not the gold-standard TTA setting. Conjugate PL seems to be sensitive to temperature tuning (e.g., in Table 1 temperature scaling is required to outperform baselines).

---

> ### Author Response · Authors · 2022-08-02
> **Response to the clarifications requested by the reviewer U1Uv : Temperature Scaling, Best TTA loss (Part 1/2)**
>
> We thank the reviewer for appreciating the novelty of our work and posting intriguing questions. Below we try to clarify the points raised by the reviewer.
>
> **Temperature Scaling :** We agree with the reviewer that the role of several heuristics in test-time adaptation is not completely understood. This includes updating only the batch norm layers as opposed to all the parameters [3]. In this work, via the meta-learning experiments, we discovered that temperature scaling is another important hyper-parameter that improves the performance of all the previous baselines as well. However, we do believe that our findings about temperature scaling are orthogonal to the contribution of conjugate pseudo-label framework. At a high level, test-time adaptation has to be appropriately regularized to prevent the updates over batches from taking the model too far: updating only a few batch norm parameters is one way to do that, and perhaps temperature scaling provides a similar beneficial regularization effect by making the network predictions on unlabeled inputs less confident. Understanding the role of these heuristics more concretely would be an interesting direction for future work.
> Additionally, our claim that conjugate PL are less sensitive to temperature scaling was based on polyloss imagenet results, but we do agree with the reviewer that this is not a global phenomenon, and so we have removed this claim entirely from the manuscript.
>
> **Regarding Validation Data :** Yes, we need validation data to tune the temperature scaling parameter, but to the best of our knowledge, all previous adaptation methods also require some form of validation data [1][2]. For example, the reviewer asked about TENT [3] – while their paper fixes a learning rate of 0.001, we found in our experiments that the performance is indeed sensitive to the choice of learning rate. So we think that our findings on temperature scaling show that it is a hyper-parameter worth tuning, along with learning rate, when doing test-time adaptation. Furthermore, temperature scaling is not necessary for using our proposed conjugate pseudo-labels. We still find performance gains over other baselines in the polyloss trained classifiers even without temperature scaling, but we can improve performance across the board via temperature scaling.
>
> **Best TTA loss :** To be clear, we used the term “best” in the sense that it is what is recovered by the meta-learning procedure. We do agree that this notion of “best” is an empirical one. We tried to reflect this in our writing (L146), and have clarified this in further detail in the revision. That said, we would like to note that because meta-leant loss is parameterized by a neural network, it hence belongs to a very powerful class of functions which potentially contains (and can learn) much more complex TTA loss functions. These could have been specific to the source model and the particular distribution shift. Our claim for conjugate loss being the “best” loss is based on the fact that it consistently recovers the learnt meta-loss (upto variance in temperature and scale) across a wide range of settings as mentioned below. Showing the optimality of conjugate loss more concretely (along with origin of temperature scaling) is one of our limitations and worth exploring in future.
>
> **Best TTA loss and various shifts :** Our experiments (Appendix A10) consistently recovered the temperature scaled convex conjugate across a range of setting we tried :
> * Datasets : CIFAR10, CIFAR100, SVHN->MNIST, SVHN->USPS.
> * Architectures : ResNet26, ResNet50.
> * Different Shifts of various intensity: Various validation noises in common corruptions dataset i.e. speckle noise, gaussian blur, saturate and spatter of various intensity (1, 2, 3, 4, 5).
>
> In Appendix A10, Figure 7 shows the learnt meta-loss when adapting from SVHN$\\rightarrow$MNIST and SVHN$\\rightarrow$USPS for a cross-entropy trained source classifier. Figure 8, 9 and 10 show the learnt meta-loss for various dataset, architectures and shifts of different intensity as mentioned above in the common corruptions dataset. Figure 11 shows the leant meta-loss when the source classifier is trained using squared loss, again recovering the quadratic loss as suggested by the conjugate formulation.  That said, we do agree that one could always construct a toy dataset where a shift is specifically added based on a pre-known optimal TTA loss.
>
> In general, we believe that our work unearths an interesting connection between the choice of TTA loss and the source classifier training. The connection between the convex conjugate of source training loss and the learnt meta-loss do lay out interesting directions which can act as a pre-cursor for future work on various aspects still open to better understanding, as even pointed out by the reviewer.

---

> > ### Author Response · Authors · 2022-08-02
> > **Response to the clarifications requested by the reviewer U1Uv : Additional experiment on SVHN -> MNIST (Part 2/2)**
> >
> > **SVHN$\\rightarrow$MNIST :** As requested by the reviewer, we performed additional experiments on SVHN $\\rightarrow$ MNIST and SVHN$\\rightarrow$USPS dataset when the source classifier is trained using cross-entropy loss. In the absence of validation data, we used a fixed learning rate of $0.01$ with Adam optimizer and T=$2$ (informed guess based on average temperature values in corruptions dataset experiments) across all the baselines.
> >
> > |    Dataset    | Temperature  | Source Error | Hard PL | Robust PL |   MEMO    | Conjugate PL (ENT) |
> > |:-------------:|:------------:|:------------:|:-------:|:---------:|:---------:|:------------------:|
> > | SVHN -> MNIST |      No      |     34.17    |  21.54  |   27.44   | **10.67** |        14.41       |
> > |               |      Yes     |     34.17    |  21.54  |   13.26   |  **9.36** |      **9.26**      |
> > |  SVHN -> USPS |      No      |     31.84    |  26.06  |   26.81   |   22.72   |      **22.57**     |
> > |               |      Yes     |     31.84    |  26.06  | **22.32** |   22.42   |      **22.27**     |
> >
> > Here again we observe that on SVHN $\\rightarrow$ MNIST benchmark, without temperature scaling, MEMO ($10.67\\%$ error) outperforms softmax-entropy ($14.41\\%$ error). However, similar to the observation in Table 1, we see that with temperature scaling, softmax-entropy minimization ($9.26\\%$ error) is able to match the performance of MEMO ($9.36\\%$ error). Further, on the SVHN $\\rightarrow$ USPS benchmark, softmax-entropy (conjugate) and MEMO perform similar even without temperature scaling.
> >
> > We have updated the manuscript accordingly with regards to the minor comments.
> >
> > We would like to thank the reviewer again, for their detailed comments on our work, identifying valid limitations and posting intriguing questions. We would be more than happy to engage in further discussion during the review period.
> >
> > [1] Evgenia Rusak, Steffen Schneider, George Pachitariu, Luisa Eck, Peter Vincent Gehler, Oliver Bringmann, Wieland Brendel, and Matthias Bethge. If your data distribution shifts, use self learning, 2022.
> > [2] Marvin Zhang, Sergey Levine, and Chelsea Finn. Memo: Test time robustness via adaptation and augmentation, 2021.
> > [3] Dequan Wang, Evan Shelhamer, Shaoteng Liu, Bruno Olshausen, and Trevor Darrell. Tent: Fully test-time adaptation by entropy minimization, 2021.

---

> > > ### Comment · Reviewer_U1Uv · 2022-08-08
> > > **Thanks for addressing my concerns**
> > >
> > > Thanks for addressing my concerns and updating the paper with additional experiments. I have increased my score.
> > >
> > > I agree that the meta-learned loss is “best”, my concern always was regarding conjugate loss recovering this “best” loss. It seems to hold true in your experiments but as the authors agreed, there could be other distribution shifts where meta-learning will still obtain the “best” loss but conjugate loss might not recover it. I agree this can be a direction for future work but should be addressed as a limitation in the paper. For example, the discussion in the limitation section can emphasize that for some distribution shifts, the following assumption from the paper might not hold (Li227): “…as long as the learnt source parameters θ is a reasonable approximation to the true optimal θ^opt on the shifted domain, self-training with the conjugate pseudo-labels provides a reasonable proxy…”

---

> > > > ### Author Response · Authors · 2022-08-08
> > > > **Thanks to the reviewer !**
> > > >
> > > > We will add a detailed discussion on the current limitations of our work and the interesting future directions (as also discussed in the response to reviewer JZZE) to the paper.
> > > >
> > > > We again thank the reviewer for their detailed feedback and for kindly increasing the score after our response.

---

### Official Review · Reviewer_so7v · 2022-07-11

**Rating:** 8
**Confidence:** 4
**Soundness:** 3 good
**Presentation:** 4 excellent
**Contribution:** 4 excellent

**Summary:**

The paper probes and evaluates the use of meta-learnt convex conjugate of the original training loss for test time adaptation to distribution shifts.
It has shown an intuition into the development of the method via a visual analysis of the numeric form of the learnt test time loss with an analytical approximation o known losses, discovering the similarity with the so-called source loss available during training.
The pseudo labels formulated  via the conjugate seem to provide a general method of computing a pseudo-label that works for losses besides the venerable CE.


**Questions:**


I did not quite follow why the method would always produce the optimal TTA loss. It is after all learnt, and operates with a non-zero risk.


**Limitations:**

No  comments on this.


**Strengths And Weaknesses:**

The reformulation of the problem within the Legendre-Fenchel duality may indeed be the first such work in this small but growing area of work.  This area would arguably grow in significance as models in use encounter non-iid conditions.  The ideas are  clearly enunciated and are therefore easy to follow. The experiments define clear baselines and use hopefully comprehensive SOTA benchmarks. The choice of the source model, i.e. a shallow ResNet is not motivated, and several small details that aid reproducibility may be added, e.g. the number of runs for obtaining the standard deviation in Table 1.

---

> ### Author Response · Authors · 2022-08-02
> **Response to the clarifications requested by the reviewer s07v**
>
> We thank the reviewer for their generous comments on our work and appreciating the test-time adaptation setup in general.
>
> We used ResNet26 and ResNet50 for CIFAR10/100 and ImageNet respectively following the previous work as mentioned in L248-L250. Table 6,7,8 in the appendix specify the exact hyper-parameters we used for all the baselines and our method, found by gridsearch whose details are mentioned in L275-L276 in the main manuscript. Standard deviation is reported across 5 independent runs. Note that we have submitted our code in supplementary and also will be open sourcing it if the work is accepted.
>
> **Best TTA loss :** To be clear, we used the term “best” in the sense that it is what is recovered by the meta-learning procedure. We do agree that this notion of “best” is an empirical one. We tried to reflect this in our writing (L146), and have clarified this in further detail in the revision. That said, we would like to note that because meta-leant loss is parameterized by a neural network, it hence belongs to a very powerful class of functions which potentially contains (and can learn) much more complex TTA loss functions. These could have been specific to the source model and the particular distribution shift. Our claim for conjugate loss being the “best” loss is based on the fact that it consistently recovers the learnt meta-loss (upto variance in temperature and scale) across a wide range of settings as mentioned in Appendix A10.
> We experimented with various datasets (CIFAR10, CIFAR100, SVHN->MNIST, SVHN->USPS), architectures (ResNet26, ResNet50) and different shifts (various validation noises in corruptions dataset) of various intensity, consistently recovering the temperature scaled convex conjugate. Showing the optimality of conjugate loss more concretely (along with origin of temperature scaling) is one of our limitations and worth exploring in future.
>
> We again thank the reviewer for carefully reviewing our work posting intriguing question. We are more than happy to engage in further discussions during the review period.

---

> > ### Comment · Reviewer_so7v · 2022-08-08
> > **Accepting answers**
> >
> > Understood.

---

### Official Review · Reviewer_M6iL · 2022-07-12

**Rating:** 7
**Confidence:** 4
**Soundness:** 4 excellent
**Presentation:** 2 fair
**Contribution:** 4 excellent

**Summary:**

Test-time training is an emerging and promising approach for building robust machine learning models to distribution shifts. As the model can not access ground-truth labels,  the most prior test-time adaptation literatures design the unsupervised loss function, e.g. entropy minimization used in TENT. However, as claimed by this paper, it is unclear what makes a good unsupervised loss in the setting of TTA.

This paper contributes to this important problem by following three means. (1) They show that the meta-learned unsupervised objective, which can be regarded as a best unsupervised loss, recover the well-known entropy minimization when the base classifier trained with cross entropy, but it recover different unsupervised loss function when using different supervised loss function. The results suggest that the best loss function differs among the choice of the supervised loss function. (2) They analyze the phenomenon from the lens of conjugated convex function, and show that it recovers the existing test-time adaptation strategies, which is similar to the objective found by meta-learning. (3) Based on the observation, they propose conjugate pseudo-labels as a method for test-time adaptation. It can be regarded as a general case of pseudo-labels, and can be used for loss function whose pseudo-labeling strategy is not trivial (e.g., recently proposed poly loss). They empirically show that the conjugate pseudo labels match the performance of entropy minimization when the source model is trained by cross-entropy loss, and outperforms existing methods when the source model is trained by Poly loss and squared loss.

**Questions:**

---
1. Regarding section 3
- (a) What parameters of networks was updated via meta-objective function? Is it entire network, or the portion of it as with TENT?
- (b) What is the exact optimization setup? What optimizer did you use? What is the learning rate, batch size, and training epoch?
- (c) How did you repeat equation 1? Were both update computed via the same batch, or different batches?
- (d) Did the difference of the above setup affect the learned objective function? In line 127, authors mentioned that the architecture change does not affect the learned objective function. I am appreciate if you can clarify what architecture did you use, and how does they consistently recovered the same objective function (e.g., does they recover the loss function with the similar scale and temperature?).
- (e) What is the prediction score in Figure 1?

---
2. Regarding section 4
- (a) How can we derive equation 2? May be it is a elemental question, but I think it makes the paper more self-contained. At least the pointer for the derivation is necessary (same for other important equations).
- (b) What is N in algorithm 1 represents for? Algorithm 1 seems to be assume that model parameters are updated for each sample, is it actually correct, or did you use the batch of data for each update?

---
3. Regarding section 4
- (a) Again, what parameters of networks was updated during test-time? Is it same for all reported methods?
- (b) Just for sure, did you treat the test-data as stream data available in online manner, or offline data? In other words, did you repeat test-time adaptation for randomly sampled batch, or did you adapt the model only once for each samples?

---
4. Question about the comments 2 and 3 in weaknesses
- (a) Can conjugate convex function perspective can explain why meta-learning recovered scaled temperature entropy, not the simple entropy?
- (b) Can you explain why some heuristics (i.e., diversity regularization in SHOT and feature alignment objective in [1,2]) works better?


[1] Kojima, Takeshi et al. “Robustifying Vision Transformer without Retraining from Scratch by Test-Time Class-Conditional Feature Alignment.” IJCAI2022
[2] Liu, Yuejiang et al. “TTT++: When Does Self-Supervised Test-Time Training Fail or Thrive?” NeurIPS (2021).

**Limitations:**

See questions.

**Strengths And Weaknesses:**

As mentioned in the summary, this paper provides several interesting insights about how to design good unsupervised loss functions by the series of experiments and theoretical analysis. This paper is well motivated, each finding is well described and interesting. Not only does the paper explain why the existing method works well, but also provide more generic forms of the similar loss function for a broader supervised loss function, called conjugated pseudo-label. Empirical validation supports the merits of the proposed method, especially the source model is trained via unusual loss function, such as poly loss and squared loss.
While I found the paper interesting, there are several unclear points from the current manuscripts.

---
<Major comments>

(1) The details of meta-learning experiments are not sufficiently presented. For example, what parameters of the networks were updated, and what optimizer, learning rate, batch size was used? It is not clear whether these factors affect the meta-learned objective or not. There is no clear explanation about what is the prediction score in Figure 1.

Regarding the meta learning experiments, I am also concerned about whether the unsupervised loss function is learned via online adaptation setup, or offline adaptation setup. In standard meta-learning literature, in my understanding, they repeat the inner loop and outer loop many times, i.e. they assume offline adaptation setup. While the approach can reveal good objective function, it might not be effective in actual test-time (online) adaptation setup, as there is the discrepancy between meta-learning and actual test-phase. For example, in online-adaptation, adaptation speed and adaptation stability might be an issue given that there are less clues about data distribution.

(2) While I like the general story of the paper and I am positive to accept the paper, I am still not convinced that the conjugated convex perspective could fully explain what makes a good unsupervised loss function in test-time adaptation setup. For example, why does the meta-learning objective recover the scaled and temperature loss function rather than the standard entropy suggested by the equation (9)? I am wondering if it is related to the assumption that the source models are sufficiently overparameterized, or the nature of test-time adaptation setup, which needs the online adaptation (and may be fast adaptation rather than other unsupervised adaptation setup).

(3) Related to the above question, the results can not explain why several prior methods are often better than TENT in practice, even when the source model is trained via cross-entropy loss. For example, [1] shows that SHOT-IM (adding diversity regularization term) often has better results than TENT. They also show that the feature alignment approach often gives better results than the simple feature modulation approach. While I understand that it is out of scope of this paper, i.e., authors do not claim that the conjugated pseudo labeling is the best possible solution, I am curious whether conjugate optimization perspective could give further insights.


(4) While it depends on readers, I found the section 4 should be more self-contained. For example, the logic behind some important equation (e.g., eq. 2, 3, 5) should be sufficiently described.

---
<Minor comments>

* In algorithm 1, no description about N.
* The index n seems to be used with different meanings around equation 6 (n represents batch size?) and in algorithm 1 (n means the index of sample or batch).
* In Table 1, Hard PL without temperature should be bold.

---

> ### Author Response · Authors · 2022-08-02
> **Response to the clarifications requested by the reviewer M6iL : Question 1, 2 and 3 (Part 1/2)**
>
> We thank the reviewer for carefully reviewing our work and appreciating the key insights and contributions of our work. Below we try to clarify some of the points raised by the reviewer.
>
> **Question 1 :** Appendix A3 discusses the meta learning experiments in further details, mentioning the architecture for the meta-objective, the exact optimization algorithm (Algorithm Box 2) and the hyper-parameters used. We have further updated Appendix A3 to specifically include the clarifications as requested by the reviewer.
> 1(a) : We update the same portion of the network (shift and bias parameters in BatchNorm layers) as with TENT.
> 1(b) : Algorithm Box 2 lists the exact optimization setup. We used Adam optimizer, with a learning rate of $0.001$, batch size of $200$ and trained for $100$ epochs.
> 1(c) : We used different batches and have corrected the manuscript.
> 1(d) : Training the above setup (Algorithm 2) to convergence repeatedly gave us similar meta-loss functions (upto variance in the scale and temperature) across various architectures, datasets and noise types (see below). We find that using different batches (rather than same batch) for inner loop (updating the classifier) and the outer loop (updating the meta-loss) gives better performance. For the meta loss architecture, we find that using a neural network with a Transformer Encoder block as the backbone gives best performance.
>
> Our meta-learning experiments (Appendix A10) consistently recovered the temperature scaled convex conjugate across a range of settings we tried :
> * Datasets : CIFAR10, CIFAR100, SVHN->MNIST, SVHN->USPS.
> * Architectures : ResNet26, ResNet50
> * Different Shifts of various intensity: Various validation noises in common corruptions dataset i.e. speckle noise, gaussian blur, saturate and spatter of various intensity (1, 2, 3, 4, 5).
>
> In Appendix A10, Figure 7 shows the learnt meta-loss when adapting from SVHN$\\rightarrow$MNIST and SVHN$\\rightarrow$USPS. Figure 8, 9 and 10 show the learnt meta-loss for various dataset, architectures and shifts of different intensity as mentioned above in the common corruptions dataset. Similarly Figure 11 shows the leant meta-loss when the source classifier is trained using squared loss, again recovering the quadratic loss as suggested by the conjugate formulation.
>
> 1(e) Prediction score refers to the output of the classifier. For a classifier trained with cross-entropy loss, prediction score is the logit whereas for squared loss, it will be the direct output (we simply avoid using the term logit for source training losses other than cross-entropy).
>
> **Online vs Offline Adaptation Setup :** In the meta learning procedure (Algorithm 2), we mimic the online adaptation setup in the inner loop. Specifically, in each epoch, in the inner loop, a sequential batch of data is used to update the classifier with the meta loss. Then we update the meta loss using some supervised surrogate loss (Equation 1) over the updated classifier’s predictions on labeled samples from the shifted data. The inner loop in each epoch would thus correspond to one online adaptation process and we repeat this until the convergence for the learnable meta-loss (i.e. multiple epochs “only” during meta-learning the loss).
>
> **Question 2 :**
> 2(a) : We would like to clarify that Equation 2 is a generic form in which various loss functions commonly used in machine learning can naturally be expressed by definition. This includes cross-entropy loss (L162), squared loss (L164), exponential loss (L567) and the poly-loss (L215). We have added a detailed derivation for expressing cross-entropy and squared loss in the form of equation 2, as well as a derivation of their convex conjugates in Appendix A1.
> 2(b) : We used batch of data for each update. N represents total number of batches. We have updated the algorithm 1 accordingly. Thanks for pointing this out.
>
> **Question 3 :**
> 3(a) : We update the same portion of the network as with TENT, i.e. the learnable scale and shift parameters of the batch normalization layers. Yes, it is the same for all the reported methods.
> 3(b) : Yes, we treat the test-data in streaming fashion and adapt the model only once for each sample (Algorithm 3 in Appendix).

---

> > ### Author Response · Authors · 2022-08-02
> > **Response to the clarifications requested by the reviewer M6iL : Question 4 (Part 2/2)**
> >
> > **Question 4 :**
> > 4(a) : We agree with the reviewer that the role of several heuristics in test-time adaptation is not completely understood. This includes updating only the batch norm layers as opposed to all the parameters [2]. In this work, via the meta-learning experiments, we discovered that temperature scaling is another important hyper-parameter that improves the performance of all the previous baselines as well. However, we do believe that our findings about temperature scaling are orthogonal to the contribution of conjugate pseudo-label framework. At a high level, test-time adaptation has to be appropriately regularized to prevent the updates over batches from taking the model too far: updating only a few batch norm parameters is one way to do that, and perhaps temperature scaling provides a similar beneficial regularization effect by making the network predictions on unlabeled inputs less confident. Understanding the role of these heuristics more concretely would be an interesting direction for future work.
> >
> > 4(b) :  In this work we considered the loss functions which (1) factorize over the test inputs independently, and (2) are functions of the predictions / logits. Examples of such loss functions include softmax-entropy, robust pseudo labeling[1], etc. The diversity regularization in SHOT-IM and feature alignment approaches considers loss functions defined across a set of samples and over intermediate representations, and hence cannot be covered by our framework. However, diversity regularization generally involves an additive loss term to the softmax-entropy minimization, and hence is complementary to our proposed conjugate loss.
> > Extending our meta-learning framework to more involved setups where we include the intermediate representations and consider learning functions over a batch of input while accounting for their interactions is worth investigating.
> >
> > We have updated the manuscript accordingly with regards to the minor comments.
> >
> > We again thank the reviewer for their detailed comments on our work, identifying valid limitations and posting intriguing questions. We are more than happy to engage in further discussions during the review period.
> >
> > [1] Evgenia Rusak, Steffen Schneider, George Pachitariu, Luisa Eck, Peter Vincent Gehler, Oliver Bringmann, Wieland Brendel, and Matthias Bethge. If your data distribution shifts, use self learning, 2022.
> > [2] Dequan Wang, Evan Shelhamer, Shaoteng Liu, Bruno Olshausen, and Trevor Darrell. Tent: Fully test-time adaptation by entropy minimization, 2021.

---

> > > ### Comment · Reviewer_M6iL · 2022-08-08
> > > **Confirm my score**
> > >
> > > Thanks for your feedback and the updated paper. The rebuttal clarify my concerns. I confirm my accept score.

---

### Official Review · Reviewer_JZZE · 2022-07-15

**Rating:** 7
**Confidence:** 4
**Soundness:** 3 good
**Presentation:** 3 good
**Contribution:** 3 good

**Summary:**

The authors propose an approach for soft-labeling test data for test-time adaptation (TTA) purposes based on "conjugate pseudolabels". Motivated by the result that metalearning the TTA objective given test time labels roughly recovers the (temperature-scaled) training objective, the authors show that if the training objective can be expressed as a conjugate function $L(x,y) = f(h(x)) - y^Th(x)$ ($x$ inputs, $y$ labels), then the optimial test time loss can be approximated by $L(y)_{conj} = -f^*(\nabla f(h(x))$, (f* the conjugate function of f), which is independent of the label, and implies the "conjugate pseudo-labels" (the usual softmax of h(x) for CE loss). TTA results on CIFAR-10-C,CIFAR-100-C, and ImageNet-C demonstrate that conjugate pseudo-labels consistently outperform other PL approaches.

**Questions:**

See S&W section.

**Limitations:**

See S&W section.

**Strengths And Weaknesses:**

Originality: 7
Quality: 7
Clarity: 7
Significance: 7

- The paper is well written and the approach and results are interesting.
- Using conjugate PLs for polyloss (novel) delivers on-par or better results than existing methods
- The conjugate PL approach is adequately developed in the paper, but could benefit significantly from a section discussing current limitations, the more general case of $L(y)_{conj}$ not independent of $y$, the dual view, and future research directions.

---

> ### Author Response · Authors · 2022-08-02
> **Limitations and Future Work as requested by the reviewer JZZE**
>
> We thank the reviewer for carefully reviewing our work and acknowledging the interesting results across datasets. Below we discuss some additional limitations and future research directions for our work as requested by the reviewer.
>
> In this work, we proposed  a general test-time adaptation loss , based on the convex conjugate formulation which in turn was motivated by the intriguing meta learning experiments. The fact that meta-learning recovers the proposed loss hints at some kind of optimality of the loss, and we can prove that for a broad set of loss functions, the proposed unsupervised conjugate loss is close to the oracle supervised loss. However, we do not yet have a complete understanding of what’s the optimal test-time adaptation method and why.
>
> Achieving good test-time adaptation generally involves several heuristics like updating only batch norm parameters (as opposed to all parameters) [2], beyond the choice of the adaptation loss itself. While our work focused on the loss, via the meta-learning experiments, we discovered that temperature scaling is another important hyper-parameter that improves the performance of all previous baselines as well. At a high level, test-time adaptation has to be appropriately regularized to prevent the updates over batches from taking the model too far: updating only a few batch norm parameters is one way to do that, and perhaps temperature scaling provides a similar beneficial regularization effect by making the network predictions on unlabeled inputs less confident. Understanding the role of these heuristics more concretely would be an interesting direction for future work.
>
> In this work we considered source training loss functions of the form which can be represented by Equation 2. A natural extension is to broaden the class of loss functions considered. This includes :
> * Studying the case when the conjugate of the source training loss function is not independent of y (as even pointed out by the reviewer). However, this might not be crucial since the current formulation covers most of the general loss functions used in machine learning.
> * Our current meta-learning framework is limited to learning a test-time adaptation loss function over the predictions of the classifier for each individual input independently. More involved setups where we include the intermediate representations and consider learning functions over a batch of input while accounting for their interactions are worth investigating.
>
> Moving on to high level extensions, a question which in general is applicable to the larger line of work around self-training is to characterize under what sort of real distributions shifts would self-training and pseudo-labeling based approaches would work. [1] provides some insights under gaussian settings.
>
> The experimental setup in this paper considered evaluating conjugate pseudo-labels for adapting to distribution shifts. But it would also be interesting to apply conjugate pseudo-labels to the general case of standard semi-supervised and self-supervised learning.
>
> We have updated the manuscript with a brief version of above discussion. We will add this detailed discussion to the additional page in the camera ready version (if the work gets accepted).
> [1] Ananya Kumar, Tengyu Ma, and Percy Liang. Understanding self-training for gradual domain adaptation, 2020.
> [2] Dequan Wang, Evan Shelhamer, Shaoteng Liu, Bruno Olshausen, and Trevor Darrell. Tent: Fully test-time adaptation by entropy minimization, 2021.

---

### Meta-Review · Area_Chair_Dt1D · 2022-08-31

**Recommendation:** Accept
**Confidence:** Certain

**Metareview:**

All reviewers agree this paper presents a novel and principled approach to test time adaptation losses. All reviewers find the paper clearly written and contributions meaningful. I suggest acceptance.

**Award:**

No

---

### Decision · Program_Chairs · 2022-09-14

Accept